# SLIM-QN: A Stochastic, Light, Momentumized Quasi-Newton Optimizer for Deep Neural Networks

## Abstract

We propose SLIM-QN, a light stochastic quasi-Newton optimizer for training large-scale deep neural networks (DNNs). SLIM-QN addresses two key barriers in existing second-order methods for large-scale DNNs: 1) the high computational cost of obtaining the Hessian matrix and its inverse in every iteration (e.g. KFAC); 2) convergence instability due to stochastic training (e.g. L-BFGS). To tackle the first challenge, SLIM-QN uses the BFGS update rule that directly approximates the Hessian inverse using past parameters and gradients, without explicitly constructing the Hessian matrix and then computing its inverse. To achieve stable convergence, SLIM-QN introduces momentum in Hessian updates together with an adaptive damping mechanism. We provide rigorous theoretical results on the convergence of SLIM-QN in a stochastic setting. We also demonstrate that SLIM-QN has much less compute and memory overhead compared to existing second-order methods. To better understand the limitations and benefits of SLIM-QN, we evaluate its performance on various datasets and network architectures. For instance on large datasets such as ImageNet, we show that SLIM-QN achieves near optimal accuracy $1.5\times$ faster when compared with SGD ($1.36\times$ faster in wall-clock time) using the same compute resources. We also show that SLIM-QN can readily be applied to other contemporary non-convolutional architectures such as Transformers.

## 1 Introduction

Second-order methods have been extensively investigated in the classical convex optimization literature. Indeed variants such as quasi-Newton methods are known to deliver faster convergence than gradient descent (GD) and achieve the best overall run time for many tasks (Gao & Goldfarb, 2019; Rodomanov & Nesterov, 2021). However, the Achilles heel of second-order methods that has impeded their wide adoption for large-scale machine learning problems is their substantial compute and memory cost, rendering them less favorable than popular stochastic first-order methods such as SGD (Saad, 1998) and its variants (Duchi et al., 2011; Kingma & Ba, 2014).

The aforementioned barriers stem from computing second-order information (loss Hessian w.r.t model parameters and its inverse), which typically dominates run-time, especially for large-scale models such as ResNet (He et al., 2016) and Vision Transformer (Dosovitskiy et al., 2020a), on large datasets such as ImageNet (Deng et al., 2009). To mitigate these issues, approximation methods have been developed to either approximate Fisher information matrix (expected Hessian matrix under negative log-likelihood loss) (Martens & Grosse, 2015; Ba et al., 2016; Pauloski et al., 2020) or directly approximate the Hessian inverse (Fletcher, 2013; Mokhtari & Ribeiro, 2015; Moritz et al., 2016; Gower et al., 2016; Gao & Goldfarb, 2018). A prominent example of the first category is KFAC (Martens & Grosse, 2015; Ba et al., 2016) which utilizes a gradient conditioner based on Fisher information and also approximates the matrix as Kronecker product of small sub-matrices, therefore simplifies matrix inversion. An example from the second category is L-BFGS (Nocedal, 1980; Liu & Nocedal, 1989) which aims to directly approximate the Hessian inverse by iterating over the past parameter and gradient changes, without explicitly constructing the Hessian matrix itself.

These methods, while to some extent alleviate the compute and memory costs of second order methods, still experience performance or convergence issues when used for training on large-scale

models. For instance, even though KFAC has faster per-iteration convergence compared to SGD, this gain is often significantly neutralized by the need to run backward passes multiple times to estimate Fisher information in each mini-batch iteration and then perform costly matrix inversion (Pauloski et al., 2020). Similarly, stochastic BFGS variants (Mokhtari & Ribeiro, 2015; Moritz et al., 2016; Gower et al., 2016) have yet to be proven efficient in training large-scale models such as ResNet in practice. A key challenge is that computationally intensive techniques are required to address convergence instability issues in this literature (Mokhtari & Ribeiro, 2015; Moritz et al., 2016; Chang et al., 2019), which unfortunately offset the compute benefits. More recently, authors in (Goldfarb et al., 2020) propose to lessen the computation burden of matrix inversion in KFAC via BFGS-like updates with promising results on simple architectures. However the efficacy of this approach is yet to be demonstrated on practical DNNs and large-scale datasets.

To simultaneously mitigate the computation and instability barriers in second-order methods, we propose SLIM-QN, a stochastic light stable BFGS-like method that achieves convergence advantages of second-order methods, while only using modest compute and memory cost compared to other techniques such as KFAC. SLIM-QN addresses the barriers in second-order methods in two ways. To reduce compute cost while maintaining fast convergence of second-order methods, SLIM-QN introduces momentum into the Hessian update. By utilizing momentum on past parameter and gradient changes, SLIM-QN smooths out the Hessian approximation without incurring costly variance reduction methods (e.g. a separate large batch size to estimate the Hessian). Furthermore, to ensure stable convergence, SLIM-QN uses an adaptive damping mechanism to adjust gradient changes so as to guarantee positive definiteness of the approximated Hessian inverse in stochastic settings. This adaptive damping scheme effectively restrains abnormal eigenvalues in the Hessian inverse and steers the optimization trajectory towards desirable directions.

SLIM-QN delivers faster convergence compared to SGD in various models and datasets. The convergence advantage is even more striking on large-scale models and datasets. Furthermore, due to its simplicity, SLIM-QN enjoys much better wall-clock convergence gains than other second-order methods such as KFAC.

In summary, our main contributions are as follows:

1. We develop SLIM-QN, a stochastic quasi-Newton algorithm targeting large-scale models, that achieves both fast and stable convergence and low computation complexity, via introducing momentum and damping into the Hessian updates.

2. We provide a rigorous analysis for SLIM-QN showing that this algorithm converges at a linear rate for stochastic optimization problems.

3. We provide complexity analysis that demonstrates that SLIM-QN is lighter than other second-order methods leading to reductions in overall wall-clock training time.

4. Finally, we carry out comprehensive evaluations on various models and datasets that show that SLIM-QN delivers faster convergence compared to SGD, especially for large datasets such as ImageNet. For instance, to reach near optimal accuracy when training ResNet-50 on ImageNet, SLIM-QN is $1.5\times$ than SGD ($1.36\times$ faster in wall clock time). Furthermore, when training Vision Transformer models, SLIM-QN also achieves faster convergence and higher accuracy.

## 2 PRELIMINARIES

In this paper we consider a typical empirical loss minimization problem of the form $\min_{\theta} \mathcal{L}(\theta, \mathcal{X}) := \frac{1}{N}\sum_{i=1}^{N} \ell(\theta, x_i)$, where $\theta$ denotes the parameters of the model to be optimized, and $\mathcal{X} = \{x_i\}_{i=1}^{N}$ are the training data where $x_i$ consists of both features and labels. The loss function is typically minimized through some variant of gradient descent, that uses the local gradients $g_t = \nabla_{\theta}\mathcal{L}(\theta_t)$ directly to update the model parameters via iterates of the form

$$\theta_{t+1} = \theta_t - \eta_t g_t, \tag{1}$$

where $\eta_t$ denotes the step size (learning rate) at iteration $t$. However, such GD updates are typically slow especially for ill-conditioned problems (Nesterov, 2003). To speed up the convergence, often second-order methods are used. In particular, Quasi-Newton (QN) methods find an approximate Hessian inverse $\hat{H}^{-1}$ to pre-condition the gradient vector and apply the following update to minimize

the loss:

$$\boldsymbol{\theta}_{t+1} = \boldsymbol{\theta}_t - \eta_t \cdot \hat{H}^{-1} \boldsymbol{g}_t.$$

In stochastic training, the gradient vector is evaluated on a mini-batch input $\mathcal{S}_t \subseteq \mathcal{X}$, namely $\boldsymbol{g}_t = \nabla_{\boldsymbol{\theta}} \mathcal{L}(\boldsymbol{\theta}_t, \mathcal{S}_t)$. If $\hat{H}^{-1}$ is the identity matrix, the update above reduces to SGD, whereas if $\hat{H}^{-1}$ is a diagonal matrix, it reduces to adaptive training algorithms such as Adagrad (Duchi et al., 2011) or Adam (Kingma & Ba, 2014). However, to incorporate more curvature information in the optimization process, it often requires approximating $\hat{H}^{-1}$ with a full symmetric matrix.

A prime challenge in QN methods is the evaluation of $\hat{H}$ and in particular its inverse. To address this challenge the well-known Broyden–Fletcher–Goldfarb–Shanno (BFGS) algorithm has been proposed. BFGS approaches the Hessian inverse as a minimization problem:

$$\min_{\hat{H}^{-1}} \quad \left\| \hat{H}^{-1} - \hat{H}_{k-1}^{-1} \right\|^2, \quad \text{s.t.} \quad \hat{H}^{-1} \cdot \boldsymbol{y}_k = \boldsymbol{s}_k, \quad \hat{H}^{-1} \text{ is symmetric},$$

where $\boldsymbol{s}_k = \boldsymbol{\theta}_k - \boldsymbol{\theta}_{k-1}$ denotes the parameter changes, and $\boldsymbol{y}_k = \boldsymbol{g}_k - \boldsymbol{g}_{k-1}$ the gradient changes in two consecutive updates [1]. Knowing $\hat{H}_{k-1}^{-1}$ from the previous update, the current $\hat{H}^{-1}$ is obtained via:

**UpdateHessian:** $\quad \hat{H}_k^{-1} = (I - \rho_k \boldsymbol{y}_k \boldsymbol{s}_k^T)^T \hat{H}_{k-1}^{-1} (I - \rho_k \boldsymbol{y}_k \boldsymbol{s}_k^T) + \rho_k \boldsymbol{s}_k \boldsymbol{s}_k^T, \quad$ (2)

where $\rho_k = \frac{1}{\boldsymbol{y}_k^T \boldsymbol{s}_k}$. Therefore, $\hat{H}^{-1}$ is constructed in an iterative manner without explicitly computing the Hessian itself. Given such an update rule, methods such as Greedy BFGS (Rodomanov & Nesterov, 2021) show $\hat{H}^{-1}$ can converge to the real Hessian at a linear rate.

In real-world problems, $\boldsymbol{\theta}$ usually consists of millions of parameters. As a result, it is infeasible to store the whole $\hat{H}_k^{-1}$ matrix with $O(|\boldsymbol{\theta}|^2)$ memory cost. To reduce memory footprint and simplify computation, $\hat{H}^{-1}$ in BFGS is stored in the form of a sequence of history vectors $\{\boldsymbol{y}_i\}$ and $\{\boldsymbol{s}_i\}$. By exploiting the Hessian update formula in equation 2, the matrix-vector product $\hat{H}_k^{-1} \cdot \boldsymbol{g}_t$ necessary to pre-condition the gradient can be replaced by a sequence of fast vector-vector products as shown in Algorithm 1. Furthermore, to limit memory and compute costs, a limited-memory version of BFGS, L-BFGS (Nocedal, 1980) is proposed that only uses the latest $M$ history vectors when approximating the Hessian inverse.

---

**Algorithm 1** Hessian-Vector in L-BFGS

**Input:** $\boldsymbol{g}_t, \{\boldsymbol{y}_i\}_{i=1}^M, \{\boldsymbol{s}_i\}_{i=1}^M$
**Output:** $\boldsymbol{g}_t$
1: **for** $i = 0, \cdots, M-1$ **do**
2: $\quad \rho_i = \boldsymbol{s}_i^T \cdot \boldsymbol{y}_i$
3: **for** $i = 0, \cdots, M-1$ **do**
4: $\quad \alpha_i = \frac{\boldsymbol{s}_{M-i-1}^T \boldsymbol{g}_t}{\rho_{M-i-1}}$
5: $\quad \boldsymbol{g}_t = \boldsymbol{g}_t - \alpha_i \cdot \boldsymbol{y}_{M-i-1}$
6: $\boldsymbol{g}_t = \hat{H}_0^{-1} \cdot \boldsymbol{g}_t \quad \triangleright \hat{H}_0^{-1} = \frac{\boldsymbol{s}_{M-1}^T \boldsymbol{y}_{M-1}}{\boldsymbol{y}_{M-1}^T \boldsymbol{y}_{M-1}} \cdot I$
7: **for** $i = 0, \cdots, M-1$ **do**
8: $\quad \beta_i = \frac{\boldsymbol{y}_i^T \boldsymbol{g}_t}{\rho_i}$
9: $\quad \boldsymbol{g}_t = \boldsymbol{g}_t + (\alpha_{M-i-1} - \beta_i) \cdot \boldsymbol{s}_i$

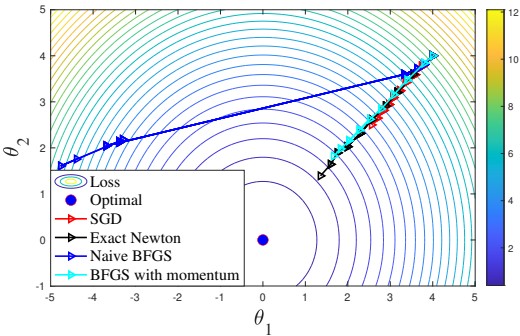

Figure 1: Optimization using SGD, Naive BFGS, BFGS with momentum and Exact Newton.

## 3 SLIM-QN

While BFGS achieves faster convergence compared to GD in full-batch training, it still suffers convergence instability in the stochastic setting, especially for large-scale DNNs. Specifically, *Naive* BFGS, which simply uses parameter $\boldsymbol{\theta}_t$ and gradients $\boldsymbol{g}_t$ at each iteration to calculate $\boldsymbol{s}_k$ and $\boldsymbol{y}_k$, suffers from severe instability due to *stochastic noise* introduced by mini-batch training (See the

---

[1]"$k$" rather than "$t$" is used in the equation as parameter/gradient used might be different from the one in equation 1

Table 1: $s_k$ and $y_k$ for Naive BFGS and for BFGS with momentum for $\mathcal{L} = \frac{1}{2} \|\boldsymbol{\theta}\|^2$

|  | Naive BFGS | BFGS with momentum |
|---|---|---|
| $\boldsymbol{s}_k$ | $\boldsymbol{\theta}_{k+1} - \boldsymbol{\theta}_k$ | $(1 - \beta_1)(\boldsymbol{\theta}_{k+1} - \boldsymbol{\theta}_k)$ |
| $\boldsymbol{y}_k$ | $\boldsymbol{\theta}_{k+1} - \boldsymbol{\theta}_k + (\boldsymbol{n}_{k+1} - \boldsymbol{n}_k)$ | $(1 - \beta_2)(\boldsymbol{\theta}_{k+1} - \boldsymbol{\theta}_k) + (1 - \beta_2)(\boldsymbol{n}_{k+1} - \boldsymbol{n}_k)$ |
| $RV$ | $\frac{2\sigma^2}{\theta_{k+1} - \theta_k}$ | $\frac{(1 - \beta_2) \cdot 2\sigma^2}{\theta_{k+1} - \theta_k}$ |

ablation study in Sec. 5.2). To stabilize the optimization, a common solution is to use a separate large batch of data when estimating $s_k$ and $y_k$ (Moritz et al., 2016; Chang et al., 2019) in order to reduce stochastic noise. However, this dramatically increases the computation cost and negates performance gains in wall-clock time.

In order to reduce variance in $\hat{H}^{-1}$ without incurring such expensive additional computation as in (Moritz et al., 2016; Chang et al., 2019), we propose a novel quasi-Newton method, SLIM-QN which introduces momentum and damping to the Hessian updates. In SLIM-QN, we first obtain the momentum of $\boldsymbol{\theta}_t$ and $\boldsymbol{g}_t$, from which we subsequently derive parameter and gradient changes $s_k$ and $y_k$. To guarantee positive definiteness of $\hat{H}^{-1}$, we further introduce adaptive damping into the update of $y_k$. With the momentumized and damped parameter and gradient changes, we can construct a consistent $\hat{H}^{-1}$ throughout the whole optimization process.

### 3.1 INTRODUCE MOMENTUM INTO THE HESSIAN UPDATE

Inspired by the success of momentum in first-order methods, we demonstrate that the Hessian update in L-BFGS can be stabilized via momentum, without requiring a large batch size or even full gradients to reduce stochastic noise. In particular, in this paper we apply momentum $\mathcal{M}_{\boldsymbol{\theta}_t}, \mathcal{M}_{\boldsymbol{g}_t}$ to past parameters $\boldsymbol{\theta}_t$ and gradients $\boldsymbol{g}_t$ during mini-batch training as follows:

$$\boldsymbol{\theta} : \mathcal{M}_{\boldsymbol{\theta}_t} = \beta_1 \cdot \mathcal{M}_{\boldsymbol{\theta}_{t-1}} + (1 - \beta_1)\boldsymbol{\theta}_t,$$
$$\boldsymbol{g} : \mathcal{M}_{\boldsymbol{g}_t} = \beta_2 \cdot \mathcal{M}_{\boldsymbol{g}_{t-1}} + (1 - \beta_2)\boldsymbol{g}_t,$$

where $\beta_1$ and $\beta_2$ are the momentum coefficients for $\boldsymbol{\theta}_t$ and $\boldsymbol{g}_t$ respectively.

Assuming that $\hat{H}^{-1}$ is updated for every $L$ mini-batch iterations, $s_k$ and $g_k$ are obtained as

$$s_k = \mathcal{M}_{\boldsymbol{\theta}_{(k+1)L}} - \mathcal{M}_{\boldsymbol{\theta}_{kL}}, \quad y_k = \mathcal{M}_{\boldsymbol{g}_{(k+1)L}} - \mathcal{M}_{\boldsymbol{g}_{kL}}.$$

This simple but effective technique works surprisingly well when gradients are noisy. To intuitively show improvements of BFGS with momentum over the naive version, we compare the stochastic optimization of a simple quadratic loss function, $\mathcal{L} = \frac{1}{2} \|\boldsymbol{\theta}\|^2$. In stochastic training, we write gradients in each mini-batch iteration as $\boldsymbol{g}_t = \boldsymbol{\theta}_t + \boldsymbol{n}_t$, where $\boldsymbol{n}_t$ denotes the *stochastic noise*. For the sake of simplicity, we model the noise as i.i.d. Gaussian, that is $\boldsymbol{n}_t \sim \mathcal{N}(0, \sigma^2)$.

Table 1 lists the expression for $s_k$ and $y_k$ for naive BFGS and for BFGS with momentum in terms of $\boldsymbol{\theta}$ and the stochastic noise. We compare the stability of $y_k$ in the two algorithms via their element-wise relative variance $RV_j = \frac{\text{Var}(\boldsymbol{y}_k(j))}{E(\boldsymbol{y}_k(j))}$, where $\boldsymbol{y}_k(j)$ denotes the $j$th element of $\boldsymbol{y}_k$. As shown in Table 1, it is easy to observe that, $RV_j^{mom.}$ using momentum is much lower than $RV_j^{naive}$ in naive BFGS. Hence, given a large momentum, SLIM-QN can significantly suppress noise and obtain a more consistent $y_k$. With less noise in $y_k$, BFGS with momentum leads to a better approximation of the real Hessian. Figure 1 visualizes the optimization trajectory for SGD, naive BFGS, BFGS with momentum and the exact Newton method. With the same initialization, BFGS with momentum is as fast as the exact Newton method, and almost twice faster than SGD. On the other hand, naive BFGS has difficulties finding the right optimization path due to the noise in $y_k$.

### 3.2 GUARANTEE POSITIVE DEFINITENESS OF THE HESSIAN VIA DAMPING

Even though adding momentum stabilizes the Hessian inverse, it cannot guarantee that the approximated Hessian is always positive definite, especially in practical stochastic and non-convex optimization. As analysed in Sec 5.2, negative or very small positive values in the Hessian spectrum

are harmful, and they immediately derail the optimization. To effectively prevent radical changes in the Hessian, SLIM-QN further introduces an adaptive damping mechanism to the Hessian update.

Positive definiteness of the Hessian hinges on stable and smooth gradient change vectors $\{\boldsymbol{y}_i\}$, especially in non-convex and stochastic settings. Hence, we choose to dampen $\boldsymbol{y}_i$ by

$$\hat{\boldsymbol{y}}_i = \tau \cdot \boldsymbol{y}_i + (1 - \tau) \cdot \boldsymbol{s}_i, \tag{3}$$

where $\hat{\boldsymbol{y}}_i$ is the damped version of $\boldsymbol{y}_i$, and $\tau$ is calculated as

$$\tau = \begin{cases} \min(\frac{1-\sigma_L}{1-\mu}, \tau_0) & \mu \leq \sigma_L < 1 \\ \min(\frac{\sigma_H - 1}{\mu - 1}, \tau_0) & \mu \geq \sigma_H > 1 \\ \tau_0 & \text{otherwise} \end{cases} \tag{4}$$

where $\mu = \frac{\boldsymbol{s}_i^T \cdot \boldsymbol{y}_i}{\boldsymbol{s}_i^T \cdot \boldsymbol{s}_i}$, $\sigma_L$ and $\sigma_H$ are the lower and upper thresholds for restraining eigenvalues in $\hat{H}^{-1}$ and $0 < \tau_0 < 1$ is a constant coefficient.

The above damping scheme prevents sudden changes of $\boldsymbol{s}_i^T \boldsymbol{y}_i$, and guarantees the smoothness of $\hat{H}^{-1}$. Equivalently, equation 3 can be considered as scaling the undamped $\hat{H}$ and adaptively shifting its spectrum by a positive constant: $\tau \cdot \hat{H} + (1 - \tau) \cdot I$. As a result, the eigenvalues of $\hat{H}$ are well controlled.

### 3.3 THE OVERALL DESCRIPTION

With momentum and damping introduced above, in this section we present the complete SLIM-QN algorithm. As shown in Algorithm 2, for each mini-batch iteration, we use $\mathcal{M}_{\boldsymbol{\theta}_t}$ and $\mathcal{M}_{\boldsymbol{g}_t}$ to accumulate momentum of parameters and gradients. For every $L$ iterations, we compute $\boldsymbol{s}_k$ and $\boldsymbol{y}_k$, apply damping, and then update the Hessian approximation. Note that in **UpdateHessian** the inverse Hessian is never explicitly computed, instead we only compute and store the history vectors $\boldsymbol{s}_k$ and $\hat{\boldsymbol{y}}_k$ that are necessary to perform the pre-conditioning step. Since at the first $2L$ iterations, $\hat{H}^{-1}$ is not ready yet, we use SGD to conduct a warmup training. After $2L$ iterations, we use $\hat{H}^{-1}$ to first pre-condition gradients $\boldsymbol{g}_t$ and then apply updates to $\boldsymbol{\theta}$. Unlike KFAC, SLIM-QN is compatible to various regularizers such as L2 and gradient regularization(Smith et al., 2021), as long as we can derive gradients from them. Finally, we add momentum to the parameter update $\Delta\boldsymbol{\theta}_t$ after pre-conditioning following the same procedure as in SGD with momentum (which is omitted in Algorithm 2).

---

**Algorithm 2** SLIM-QN algorithm

---

1: **for** $t = 1, \cdots, max\_iter$ **do**
2:     Randomly choose mini-batch input $\mathcal{S}_t \in \mathcal{X}$
3:     Compute gradients $\boldsymbol{g}_t$ given $\mathcal{S}_t$(Forward/Backward)
4:     Add weight decay: $\boldsymbol{g}_t = \boldsymbol{g}_t + wd \cdot \boldsymbol{\theta}_t$
5:     Compute momentum on $\boldsymbol{\theta}$: $\mathcal{M}_{\boldsymbol{\theta}_t} = \beta_1 \cdot \mathcal{M}_{\boldsymbol{\theta}_{t-1}} + (1 - \beta_1) \cdot \boldsymbol{\theta}_t$        $\triangleright \mathcal{M}_{\boldsymbol{\theta}_0} = \boldsymbol{\theta}_0$
6:     Compute momentum on $\boldsymbol{g}$: $\mathcal{M}_{\boldsymbol{g}_t} = \beta_2 \cdot \mathcal{M}_{\boldsymbol{g}_{t-1}} + (1 - \beta_2) \cdot \boldsymbol{g}_t$        $\triangleright \mathcal{M}_{\boldsymbol{g}_0} = \boldsymbol{g}_0$
7:     **if** $t \leq 2L$ **then**
8:         Warmup: $\boldsymbol{\theta}_{t+1} = \boldsymbol{\theta}_t - \eta_t \cdot \boldsymbol{g}_t$
9:     **else**
10:        Pre-condition: $\Delta\boldsymbol{\theta}_t = \hat{H}_k^{-1} \cdot \boldsymbol{g}_t$             $\triangleright$ Algorithm 1
11:        Update: $\boldsymbol{\theta}_{t+1} = \boldsymbol{\theta}_t - \eta_t \cdot \Delta\boldsymbol{\theta}_t$
12:     **if** $t\%L == 0$ and $t > L$ **then**
13:        $k = k + 1$
14:        $\boldsymbol{s}$: $\boldsymbol{s}_k = \mathcal{M}_{\boldsymbol{\theta}_t} - \mathcal{M}_{\boldsymbol{\theta}_{t-L}}$
15:        $\boldsymbol{y}$: $\boldsymbol{y}_k = \mathcal{M}_{\boldsymbol{g}_t} - \mathcal{M}_{\boldsymbol{g}_{t-L}}$
16:        Damping: $\hat{\boldsymbol{y}}_k = \tau \cdot \boldsymbol{y}_k + (1 - \tau) \cdot \boldsymbol{s}_k$        $\triangleright \tau$ from equation 4
17:        $\hat{H}^{-1}$: $\hat{H}_k^{-1} = $ **UpdateHessian**$(\hat{H}_{k-1}^{-1}, \boldsymbol{s}_k, \hat{\boldsymbol{y}}_k)$    $\triangleright$ equation 2, not explicitly computed

---

# 4 THEORETICAL GUARANTEES FOR SLIM-QN

In this section, we first present convergence guarantees for SLIM-QN, and then discuss the compute and memory costs for SGD, KFAC, and SLIM-QN.

## 4.1 CONVERGENCE GUARANTEES

Following the framework of quasi-Newton method in Wang et al. (2017) in stochastic optimization, we prove that SLIM-QN converges to the optimum at a linear rate, under proper assumptions as follows:

**AS 1.** $\mathcal{L}(\boldsymbol{\theta})$ *is $\lambda$-PL in that it satisfies Polyak-Lojasiewicz (PL) condition for a constant $\lambda > 0$:* $\|\nabla\mathcal{L}(\boldsymbol{\theta})\|^2 \geq \lambda\mathcal{L}(\boldsymbol{\theta})$.

**AS 2.** $\ell_i(\boldsymbol{\theta})$ *is $\Lambda$-smooth for $1 \leq i \leq N$, $\Lambda > 0$: $\forall \boldsymbol{\theta}_1, \boldsymbol{\theta}_2, \ell_i(\boldsymbol{\theta}_2) \leq \ell_i(\boldsymbol{\theta}_1) + \langle\nabla\ell_i(\boldsymbol{\theta}_1), \boldsymbol{\theta}_2 - \boldsymbol{\theta}_1\rangle + \frac{\Lambda}{2}\|\boldsymbol{\theta}_2 - \boldsymbol{\theta}_1\|^2$.*

**AS 3.** *For every sequence $\boldsymbol{\theta}_1, \boldsymbol{\theta}_2, \cdots$ such that $\lim_{t\to\infty}\mathcal{L}(\boldsymbol{\theta}_t) = 0$, then for all $1 \leq i \leq N$, $\lim_{t\to\infty}\ell_i(\boldsymbol{\theta}_t) = 0$*

We note that compared to typical strong convexity assumptions the PL condition in AS 1 (Polyak, 1963) applies to much broader settings including when the loss is nonconvex as it only requires lower bounded variance in gradients, rather than strict positive definiteness required for the Hessian with strong convexity. AS 3 assumes that the global minima of the summands $\ell_i$ are the same as the global minima of their sum. This is in line with what has been observed in over-parameterized deep learning models (Ma et al., 2018).

Before we state our main result we require several auxiliary lemmas. First Lemma 1 states that $\boldsymbol{s}_i^T \cdot \hat{\boldsymbol{y}}_i$ always lies in $[\sigma_L, \sigma_H] \cdot \boldsymbol{s}_i^T \boldsymbol{s}_i$. With Lemma 1 in place, in Lemma 2 we bound the approximation of the Hessian $\hat{H}_k^{-1}$ at $k$th Hessian update. Lemma 3 further establishes smoothness on $\mathcal{L}(\boldsymbol{\theta})$ given each $\ell_i$ is smooth. While Lemma 4 further bounds gradient variance in mini-batch training.

**Lemma 1.** *Given damping scheme in equation 3, if we choose $\tau$ according to equation 4, then $\sigma_L \leq \frac{\boldsymbol{s}_i^T \cdot \hat{\boldsymbol{y}}_i}{\boldsymbol{s}_i^T \cdot \boldsymbol{s}_i} \leq \sigma_H$.*

**Lemma 2.** *Given Lemma 1, at the $k$-th Hessian update, $\hat{H}_k^{-1}$ during the optimization is bounded by $\xi I \preceq \hat{H}_k^{-1} \preceq \Xi I$, where $\Xi = (M+1)\frac{1}{\sigma_L}$, and $\xi = \frac{1}{\sigma_H}$. $\sigma_L$ and $\sigma_H$ is the lower and upper bound for $\frac{\boldsymbol{s}_i^T \cdot \hat{\boldsymbol{y}}_i}{\boldsymbol{s}_i^T \cdot \boldsymbol{s}_i}$ in Lemma 1.*

**Lemma 3.** *Assume AS 2 holds, then loss function $\mathcal{L}(\boldsymbol{\theta})$ is at least $\Lambda$-smooth.*

**Lemma 4.** *Assume AS 2-3 holds, with Lemma 3, at iteration $t$ with mini-batch input $\mathcal{S}_t$, where each sample is randomly sampled from $\mathcal{X}$ with replacement, gradient $\nabla\mathcal{L}(\boldsymbol{\theta}_t; \mathcal{S}_t)$ satisfies*

$$E_{\mathcal{S}_t}[\|\nabla\mathcal{L}(\boldsymbol{\theta}_t; \mathcal{S}_t)\|^2] \leq 2\Lambda \cdot \mathcal{L}(\boldsymbol{\theta}_t)$$

With $\hat{H}_k^{-1}$ and gradient variance bounded, we can derive the following convergence theorem.

**Theorem 1.** *Assume AS 1-3 hold at each iteration $t$ of SLIM-QN with mini-batch input $\mathcal{S}_t$ where each sample is randomly sampled from $\mathcal{X}$ with replacement, then the expectation of $\mathcal{L}(\boldsymbol{\theta}_t)$ satisfies*

$$E_{\mathcal{S}_t}[\mathcal{L}(\boldsymbol{\theta}_t)] \leq \alpha_{t-1} E_{\mathcal{S}_{t-1}}[\mathcal{L}(\boldsymbol{\theta}_{t-1})],$$

*where $\alpha_{t-1} = 1 - \eta_{t-1}\lambda\xi + \eta_{t-1}^2\Lambda^2\Xi^2$.*

Proofs for the lemmas and theorem are provided in the appendix.

**Remark 1.** *By choosing $\eta_{t-1}$ such that $\alpha_{t-1} < 1$, SLIM-QN converges at a linear rate.*

**Remark 2.** *We note that Theorem 1 also applies to convex settings, as strong convexity implies $\|\nabla\mathcal{L}(\boldsymbol{\theta})\|^2$ is lower bounded by $\mathcal{L}(\boldsymbol{\theta})$ for an appropriate $\lambda > 0$ and hence AS 1 holds.*

Table 2: Computations and Memory in SGD, KFAC, and SLIM-QN

| | SGD | KFAC | sL-BFGS | SLIM-QN |
|---|---|---|---|---|
| | | Computation | | |
| Fwd&Bwd | $bC_{\text{fb}}$ | $\alpha_1\|\boldsymbol{\theta}\| + \gamma bC_{\text{fb}} + \frac{1}{L}\sum(d_i^3 + (\frac{\|\boldsymbol{\theta}_i\|}{d_i})^3)$ | $\alpha_2\|\boldsymbol{\theta}\| + 2bC_{\text{fb}} + \frac{1}{L}b_H C_{\text{fb}}$ | $bC_{\text{fb}} + \alpha_3\|\boldsymbol{\theta}\|$ |
| Opt | $\alpha_0\|\boldsymbol{\theta}\|$ | $\alpha_0\|\boldsymbol{\theta}\| + 2\sum(d_i + \frac{\|\boldsymbol{\theta}_i\|}{d_i})\|\boldsymbol{\theta}_i\|$ | $\alpha_0\|\boldsymbol{\theta}\| + 2M\|\boldsymbol{\theta}\|$ | $\alpha_0\|\boldsymbol{\theta}\| + 2M\|\boldsymbol{\theta}\|$ |
| | | Memory | | |
| Fwd&Bwd | $bM_{\text{fb}}$ | $bM_{\text{fb}} + \beta_1\|\boldsymbol{\theta}\|$ | $bM_{\text{fb}} + \beta_2\|\boldsymbol{\theta}\| + \frac{1}{L}b_H M_{\text{fb}}$ | $bM_{\text{fb}} + \beta_3\|\boldsymbol{\theta}\|$ |
| Opt | $\beta_0\|\boldsymbol{\theta}\|$ | $\beta_0\|\boldsymbol{\theta}\| + 2\sum(d_i^2 + (\frac{\|\boldsymbol{\theta}_i\|}{d_i})^2)$ | $\beta_0\|\boldsymbol{\theta}\| + 2M\|\boldsymbol{\theta}\|$ | $\beta_0\|\boldsymbol{\theta}\| + 2M\|\boldsymbol{\theta}\|$ |

- $d_i$: input dim in layer $i$. $b$: batch size. $b_H$: batch size for the Hessian approx. $\|\boldsymbol{\theta}_i\|$: #params in layer $i$.

## 4.2 COMPUTE AND MEMORY COST

As mentioned before, SLIM-QN aims to reduce the complexity of approximating the Hessian. In this section, we summarize the compute and memory cost of SGD, KFAC, stochastic L-BFGS (sL-BFGS) variants(Chang et al., 2019) and SLIM-QN, and demonstrate the cost advantage of SLIM-QN.

Given a model with parameter $\boldsymbol{\theta}$, we use $C_{\text{fb}}$ and $M_{\text{fb}}$ to represent the compute and memory cost of a forward/backward (Fwd/Bwd) pass with a batch size of $b = 1$. Furthermore, $C_{\text{opt}}$ denotes the compute cost of model updates (Opt) which consists of gradient reduction, computing the update $\Delta\boldsymbol{\theta}$, and applying it to $\boldsymbol{\theta}$.

Table 2 summarizes the compute and memory cost of SGD, KFAC, sL-BFGS and SLIM-QN. Compared to SGD, during the forward and backward passes, SLIM-QN needs to additionally compute $\mathcal{M}_{\boldsymbol{\theta}}, \mathcal{M}_{\boldsymbol{g}}$, for which the complexity increases linearly with model size ($\alpha_2\|\boldsymbol{\theta}\|$). The main extra compute SLIM-QN introduces is the Hessian-vector product, in which we need to iterate over $\{\boldsymbol{s}_i\}_{i=1}^M$ and $\{\boldsymbol{y}_i\}_{i=1}^M$, as shown in Algorithm 1. The complexity increases linearly with the number of stored history vectors and model size ($2M\|\boldsymbol{\theta}\|$). While, compared to $O(bC_{\text{fb}})$ complexity in forward and backward pass, such operations add relatively marginal cost.

As a comparison, KFAC though only approximates diagonal blocks of the Fisher matrix, it still adds significant additional computations through 1) multiple backward passes to update factors ($\gamma bC_{\text{fb}}$ with $\gamma \geq 1$), 2) matrix inversion ($\sum(d_i^3 + (\frac{\|\boldsymbol{\theta}_i\|}{d_i})^3)$) for every L iterations, and 3) Matrix-vector products ($2\sum(d_i + \frac{\|\boldsymbol{\theta}_i\|}{d_i})\|\boldsymbol{\theta}_i\|$). If the Fisher matrix is updated more frequently (that is for small $L$), then the amortized cost for matrix inversion is even more striking. On the other hand, sL-BFGS also resorts to computation-intensive operations including full-batch gradients and a separate large batch to estimate the Hessian, which respectively adds amortized costs of $O(bC_{\text{fb}})$ and $\frac{1}{L}b_H C_{\text{fb}}$.

As for memory usage, compared to SGD, SLIM-QN mainly needs $O(2M\|\boldsymbol{\theta}\|)$ to store history vectors $\{\boldsymbol{s}_i\}_{i=1}^M$ and $\{\boldsymbol{y}_i\}_{i=1}^M$. sL-BFGS needs the same storage for $\boldsymbol{s}_i, \boldsymbol{y}_i$, and amortized cost of $O(\frac{1}{L}b_H M_{\text{fb}})$ for additional backward passes. While KFAC needs $O(2\sum(d_i^2 + (\frac{\|\boldsymbol{\theta}_i\|}{d_i})^2))$ to store sub-matrices and their inverse, where the actual memory footprint hinges on model architectures. In practice, $M$ is set to be $10 \sim 20$, which ensures that memory usage is manageable in SLIM-QN.

## 5 EMPIRICAL ANALYSES

We conduct various experiments on computer vision (CV) problems, where SGD has been widely used. Methods like Adam (Kingma & Ba, 2014) and AdaGrad (Duchi et al., 2011) greatly under-perform SGD (Defazio & Jelassi, 2021). Two metrics are used to evaluate the performance: iteration-wise convergence and wall-clock convergence. The iteration-wise metric shows the pure convergence promises of the optimizer; while the wall-clock convergence further captures the impacts of computation complexity on the run-time. Furthermore, we also conduct an ablation study that investigates how the components in SLIM-QN (momentum and damping) affect the optimization process.

The current implementation in PyTorch (Paszke et al., 2019) supports various DNN models in a multi-GPU system. During training, after gradients are synchronized across GPUs, we keep updating the momentum, $\mathcal{M}_{\boldsymbol{\theta}_t}$ and $\mathcal{M}_{\boldsymbol{g}_t}$ on each GPU. As a result, each GPU stores a copy of the Hessian

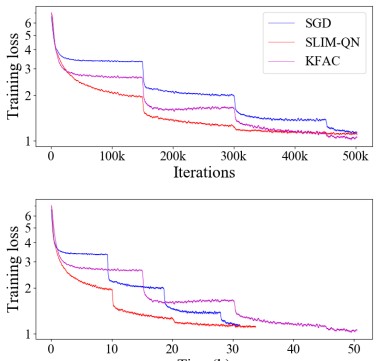 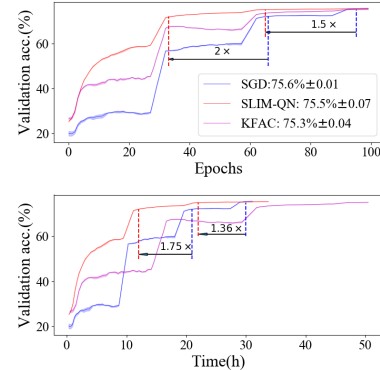

Figure 2: Training loss (left) and validation accuracy (right) of SLIM-QN, KFAC and SGD on ImageNet using ResNet-50 model. The model trained with SLIM-QN benefits from faster early-stage convergence and achieve comparable generalization performance as SGD. We plot the mean and standard error over 3 runs with different random seeds.

inverse and locally performs gradient conditioning without communicating across GPUs. We test models: ResNet18, ResNet50, Vision Transformer on datasets: CIFAR-10(Krizhevsky et al., 2009) and ImageNet(Deng et al., 2009). We also run SGD and KFAC as two baselines for comparison.

## 5.1 EXPERIMENTS ON IMAGENET CLASSIFICATION

ImageNet classification has been the gold standard for evaluating performance of different optimization algorithms for CV models. Compared to CIFAR-10, ImageNet consists of much more training and test images ($\sim$1.2M training and $\sim$50K test images), categorized into 1000 classes. Therefore, convergence on ImageNet can better reveal the optimizer's promises in practical problems.

During data pre-processing, we resize images to $256 \times 256$, and randomly crop to $224 \times 224$, and then randomly flip each image. Each image is normalized using pre-computed mean and variance.

### 5.1.1 RESNET-50

Figure 2 shows iteration-wise (Top) and wall-clock (Bottom) convergence on ResNet-50 using SGD, KFAC and SLIM-QN. Detailed hyper-parameter settings are provided in the appendix. SLIM-QN enjoys very fast per-iteration convergence, and reaches near optimal accuracy $1.5\times$ faster than SGD, and even $2\times$ faster in the early-stages. Furthermore, it also generalizes well on the validation set, and finally reaches comparable validation accuracy to SGD.

The benefit of SLIM-QN is even more striking in terms of wall-clock time. Due to light compute costs, it is $1.75 \times /1.36\times$ faster in the early and late stages compared to SGD. Whereas in KFAC, the wall-clock performance is significantly neutralized by its additional compute costs.

### 5.1.2 VISION TRANSFORMER

As a step towards understanding the efficacy of second-order optimizers on contemporary Transformer-based CV models, we perform experiments on a small ($10M$ parameters) Vision Transformer (ViT) (Dosovitskiy et al., 2020b) using SLIM-QN on ImageNet. Details on hyperparameters, model architecture and experiments on further datasets are deferred to the Appendix.

As shown in Fig. 3, SLIM-QN benefits from faster early-stage convergence compared to the SGD, which is consistent with our findings for ResNet. Furthermore, we observe that SLIM-QN finds a solution with good generalization achieving a final validation accuracy slightly higher than SGD.

## 5.2 ABLATION STUDY: THE EFFECTS OF MOMENTUM AND DAMPING

In this section, we give more insight into the effects of momentum and damping used in SLIM-QN. To this end, we ablate two critical components in SLIM-QN: momentum and damping in the Hessian

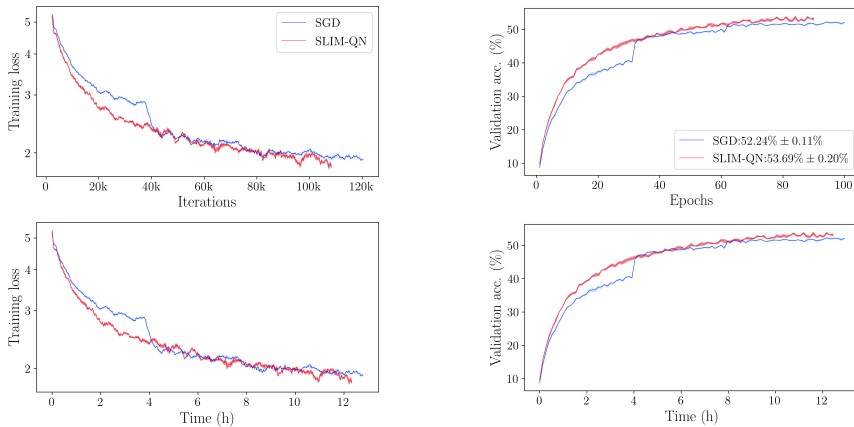

Figure 3: Training loss (left) and validation accuracy (right) of SLIM-QN and SGD on ImageNet using a Vision Transformer model. The model trained with SLIM-QN benefits from faster early-stage convergence and achieve better generalization performance compared to SGD. We plot the mean and standard error over 3 runs with different random seeds.

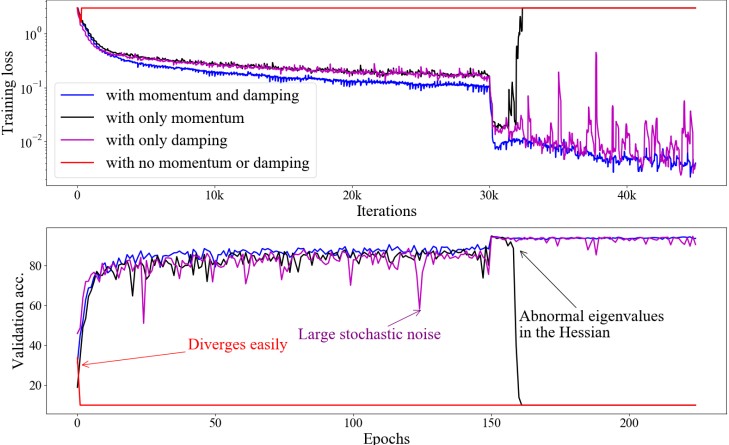

Figure 4: Ablation analysis for SLIM-QN on ResNet-18/CIFAR-10 (batch size: 256).

approximation, and then use the ablated version to train ResNet-18 on CIFAR-10. We focus on CIFAR-10 since we observed more convergence instability on this dataset compared to others.

Figure 4 shows convergence using the ablated SLIM-QN with only momentum (black), with only damping (purple), and with no momentum or damping (red). Due to stochastic noise, the ablated version of SLIM-QN without momentum or damping diverges easily in the early stages. With momentum (black), the whole optimization is significantly stabilized. However, it still fails to converge when there is an abrupt change in the loss landscape (for example, when learning rate decays). With damping (purple), the Hessian approximation is effectively restrained, especially when such sudden changes in the loss landscape happen. It is interesting to observe that while damping prevents divergence, the whole training is still largely affected by stochastic noise. Notable fluctuation in the loss and accuracy is commonly observed during training. As a comparison, the complete SLIM-QN (blue) effectively addresses these issues achieving much more stable convergence.

## 6 CONCLUSION

In this paper, we propose SLIM-QN, a quasi-Newton method that simultaneously mitigates computation and convergence instability barriers in second-order methods. SLIM-QN introduces momentum and damping into the Hessian update, which obviates the need for estimating the Hessian with high costs. Empirical analyses on CV models, such as ResNet-50 and Vision Transformer show that SLIM-QN achieves faster convergence in the early stages, and reaches comparable accuracy to SGD.

## REPRODUCIBILITY STATEMENT

Implementation of SLIM-QN in a multi-GPU platform is described at the beginning of Sec 5. A copy of code is included in supplementary materials. Appendix B lists all hyperparameters for obtaining results on ImageNet using ResNet-50 and ViT models. Moreover, Appendix C presents more results on CIFAR-10 using ResNet-18 and ViT models. We also describe some tips of tuning hyperparameters in Appendix F.

For theoretical results, proofs of all lemmas and theorems are provided in Appendix A.

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

# A PROOFS OF LEMMAS AND THEOREMS

This appendix are organized as follows:

1. Section A.1-A.4 presents the proof of Lemma 1-4.
2. Section A.5 presents the proof of Theorem 1.

## A.1 PROOF OF LEMMA 1

*Proof.* According to equation 3, $s_i^T \hat{y}_i = s_i^T(\tau y_i + (1-\tau)s_i) = (\mu\tau + 1 - \tau)s_i^T s_i$, where $\mu = \frac{s_i^T y_i}{s_i^T s_i}$.

For $\mu \leq \sigma_L$, two cases need to be considered: 1) $\tau = \tau_0$; 2) $\tau = \frac{1-\sigma_L}{1-\mu}$.

If $\tau = \tau_0$, then $\frac{1-\sigma_L}{1-\mu} \geq \tau_0$, and $\mu\tau + 1 - \tau \geq \sigma_L$ If $\tau = \frac{1-\sigma_L}{1-\mu}$, then $\mu\tau + 1 - \tau = \sigma_L$

Therefore, when $\mu \leq \sigma_L$, $s_i^T \hat{y}_i \geq \sigma_L s_i^T s_i$

For $\sigma_L < \mu < \sigma_H$:

We can write $\mu\tau + 1 - \tau = \mu\tau_0 + 1 - \tau_0$. It is easy to show that $\mu\tau_0 + 1 - \tau_0 - \sigma_L \geq (1-\sigma_L)(1-\tau_0) > 0$ and $\mu\tau_0 + 1 - \tau_0 - \sigma_H \leq (1-\sigma_H)(1-\tau_o) < 0$.

Therefore, when $\sigma_L < \mu < \sigma_H$, $\sigma_L s_i^T s_i < s_i^T \hat{y}_i < \sigma_H s_i^T s_i$.

For $\mu \geq \sigma_H$, similarly two cases might arise: 1) $\tau = \tau_0$; 2) $\tau = \frac{\sigma_H - 1}{\mu - 1}$.

If $\tau = \tau_0$, then $\frac{\sigma_H - 1}{\mu - 1} \geq \tau_0$, and $\mu\tau + 1 - \tau \leq \sigma_H$. If $\tau = \frac{\sigma_H - 1}{\mu - 1}$, then $\mu\tau + 1 - \tau = \sigma_H$.

Therefore, when $\mu \geq \sigma_H$, $s_i^T \hat{y}_i \leq \sigma_H s_i^T s_i$

In summary, $\sigma_L \leq \frac{s_i^T \cdot \hat{y}_i}{s_i^T \cdot s_i} \leq \sigma_H$. $\qquad\square$

## A.2 PROOF OF LEMMA 2

*Proof.* **Lower bound**: $\hat{H}^{-1}$ is initialized as $\hat{H}_0^{-1} = \frac{s_0^T \hat{y}_0}{\hat{y}_0^T \hat{y}_0} \cdot I$. According to Lemma 1, there exists $H_0(\theta) \preceq \sigma_H I$ such that $\hat{y}_0 = H_0 \cdot s_0$.

Therefore, $\frac{s_0^T \hat{y}_0}{\hat{y}_0^T \hat{y}_0} \cdot I = \frac{s_0^T H_0 s_0}{s_0^T H_0 \cdot H_0 s_0} \cdot I = \frac{(s_0^T H_0^{1/2})(H_0^{1/2} s_0)}{(s_0^T H_0^{1/2}) \cdot H_0 \cdot (H_0^{1/2} s_0)} \cdot I \succeq \frac{1}{\sigma_H} I$.

Then for $k \geq 1$, assuming $\hat{H}_{k-1}^{-1} \succeq \frac{1}{\sigma_H} I$ hold, based on equation 2, $\hat{H}_k^{-1} = (I - \rho_k \hat{y}_k s_k^T)^T \hat{H}_{k-1}^{-1}(I - \rho_k \hat{y}_k s_k^T) + \frac{s_k s_k^T}{s_k^T \hat{y}_k}$.

Because $(I - \rho_k \hat{y}_k s_k^T)^T \hat{H}_{k-1}^{-1}(I - \rho_k \hat{y}_k s_k^T)$ is positive definite, we can bound $\hat{H}_k^{-1}$ as: $\hat{H}_k^{-1} \succeq \frac{s_k s_k^T}{s_k^T \hat{y}_k} = \frac{s_k s_k^T}{s_k^T \cdot H_k \cdot s_k} \succeq \frac{1}{\sigma_H} I$

Therefore, lower bound of $\hat{H}_k^{-1}$, $\xi = \frac{1}{\sigma_H}$.

**Upper bound**: Since $H_0(\theta) \succeq \sigma_L$, we can get $\frac{s_0^T \hat{y}_0}{\hat{y}_0^T \hat{y}_0} \cdot I = \frac{(s_0^T H_0^{1/2})(H_0^{1/2} s_0)}{(s_0^T H_0^{1/2}) \cdot H_0 \cdot (H_0^{1/2} s_0)} \cdot I \preceq \frac{1}{\sigma_L}$.

Similarly, for $k \geq 1$, we assume $\hat{H}_{k-1}^{-1} \preceq k \frac{1}{\sigma_L}$ hold.

For the first part in equation 2, let $Q = (I - \rho_k \hat{y}_k s_k^T)$. Then for $\forall x \neq 0$, $x^T \cdot Q^T \hat{H}_{k-1}^{-1} Q \cdot x = (Qx)^T \cdot \hat{H}_{k-1}^{-1} \cdot (Qx) \leq \frac{k}{\sigma_L} x^T \cdot (Q^T Q) \cdot x$.

Let $P = \frac{s_k \hat{y}_k^T \hat{y}_k s_k^T}{s_k^T \hat{y}_k s_k^T \hat{y}_k} - \frac{\hat{y}_k s_k^T + s_k \hat{y}_k^T}{s_k^T \hat{y}_k}$, then $Q^T Q = I + P$.

Because $P$ is rank-1 matrix with eigenvalue $-1$ and eigenvector $\hat{\boldsymbol{y}}_k$, we have $\boldsymbol{x}^T \cdot Q^T \hat{H}_{k-1}^{-1} Q \cdot \boldsymbol{x} \leq \frac{k}{\sigma_L} \boldsymbol{x}^T \cdot (I + P) \cdot \boldsymbol{x} \leq \frac{k}{\sigma_L} \boldsymbol{x}^T \boldsymbol{x}$

For the second part in equation 2, we can directly get $\frac{\boldsymbol{s}_k \boldsymbol{s}_k^T}{\boldsymbol{s}^T \boldsymbol{y}_k} = \frac{\boldsymbol{s}_k \boldsymbol{s}_k^T}{\boldsymbol{s}_k^T \cdot H_k \cdot \boldsymbol{s}_k} \preceq \frac{1}{\sigma_L} I$.

Therefore, $\boldsymbol{x}^T \cdot \hat{H}_k^{-1} \cdot \boldsymbol{x} \leq \frac{k}{\sigma_L} \boldsymbol{x}^T \boldsymbol{x} + \frac{1}{\sigma_L} \boldsymbol{x}^T \boldsymbol{x} = \frac{k+1}{\sigma_L} \boldsymbol{x}^T \boldsymbol{x}$

In SLIM-QN, $k$ is at most $M$, which is the length of history vector, therefore $\Xi = (M+1)\frac{1}{\sigma_L}$.

In summary, $\frac{1}{\sigma_H} I \preceq \hat{H}_k^{-1} \preceq (M+1)\frac{1}{\sigma_L} I$ □

## A.3 PROOF OF LEMMA 3

*Proof.* Given AS 2 hold, we have

$$\|\nabla \ell_i(\boldsymbol{\theta}_1) - \nabla \ell_i(\boldsymbol{\theta}_2)\| \leq \Lambda \|\boldsymbol{\theta}_1 - \boldsymbol{\theta}_2\|.$$

For $\mathcal{L}$, we have

$$\nabla \mathcal{L}(\boldsymbol{\theta}_1) - \nabla \mathcal{L}(\boldsymbol{\theta}_2) = \frac{1}{N} \sum_{i=1}^{N} \nabla \ell_i(\boldsymbol{\theta}_1) - \nabla \ell_i(\boldsymbol{\theta}_2)$$

Then,

$$\|\nabla \mathcal{L}(\boldsymbol{\theta}_1) - \nabla \mathcal{L}(\boldsymbol{\theta}_2)\| = \frac{1}{N} \left\| \sum_{i=1}^{N} \nabla \ell_i(\boldsymbol{\theta}_1) - \nabla \ell_i(\boldsymbol{\theta}_2) \right\|$$

$$\overset{\text{TI}}{\leq} \frac{1}{N} \sum_{i=1}^{N} \|\nabla \ell_i(\boldsymbol{\theta}_1) - \nabla \ell_i(\boldsymbol{\theta}_2)\|$$

$$\leq \frac{1}{N} \sum_{i=1}^{N} \Lambda \|\boldsymbol{\theta}_1 - \boldsymbol{\theta}_2\|$$

$$= \Lambda \|\boldsymbol{\theta}_1 - \boldsymbol{\theta}_2\|$$

"TI" indicates triangle inequality. Therefore, $\mathcal{L}$ is at least $\Lambda$-smooth. □

## A.4 PROOF OF LEMMA 4

*Proof.* $E_{\mathcal{S}_t}[\|\nabla \mathcal{L}(\boldsymbol{\theta}_t; \mathcal{S}_t)\|^2] = E_{\mathcal{S}_t}[\langle \frac{1}{b} \nabla \sum_{i=1}^{b} \ell_i(\boldsymbol{\theta}_t), \frac{1}{b} \nabla \sum_{i=1}^{b} \ell_i(\boldsymbol{\theta}_t) \rangle]$

Expand summation and regroup,

$E_{\mathcal{S}_t}[\|\nabla \mathcal{L}(\boldsymbol{\theta}_t; \mathcal{S}_t)\|^2] = E_{\mathcal{S}_t}[\frac{1}{b^2} \sum_{i=1}^{b} \|\nabla \ell_i(\boldsymbol{\theta}_t)\|^2 + \frac{1}{b^2} \sum_{i=1}^{b} \sum_{j=1, \neq i}^{b} \langle \nabla \ell_i(\boldsymbol{\theta}_t), \nabla \ell_j(\boldsymbol{\theta}_t) \rangle]$

Take expectation on each sample,

$E_{\mathcal{S}_t}[\|\nabla \mathcal{L}(\boldsymbol{\theta}_t; \mathcal{S}_t)\|^2] = \frac{1}{b^2} \sum_{i=1}^{b} E_{\boldsymbol{x}_i}[\|\nabla \ell_i(\boldsymbol{\theta}_t)\|^2] + \frac{1}{b^2} \sum_{i=1}^{b} \sum_{j=1, \neq i}^{b} E_{\boldsymbol{x}_i, \boldsymbol{x}_j}[\langle \nabla \ell_i(\boldsymbol{\theta}_t), \nabla \ell_j(\boldsymbol{\theta}_t) \rangle]$

Because $\boldsymbol{x}_i$ and $\boldsymbol{x}_j$ are independent, the second part can be simplified as,

$E_{\mathcal{S}_t}[\|\nabla \mathcal{L}(\boldsymbol{\theta}_t; \mathcal{S}_t)\|^2] = \frac{1}{b^2} \sum_{i=1}^{b} E_{\boldsymbol{x}_i}[\|\nabla \ell_i(\boldsymbol{\theta}_t)\|^2] + \frac{1}{b^2} \sum_{i=1}^{b} \sum_{j=1, \neq i}^{b} \|\nabla \mathcal{L}(\boldsymbol{\theta}_t)\|^2$

With further simplification, we get

$E_{\mathcal{S}_t}[\|\nabla \mathcal{L}(\boldsymbol{\theta}_t; \mathcal{S}_t)\|^2] = \frac{1}{b^2} \sum_{i=1}^{b} E_{\boldsymbol{x}_i}[\|\nabla \ell_i(\boldsymbol{\theta}_t)\|^2] + \frac{b-1}{b} \|\nabla \mathcal{L}(\boldsymbol{\theta}_t)\|^2$

Given AS 2 and Lemma 3, we have

$\|\nabla \ell_i(\boldsymbol{\theta}_t)\|^2 \leq 2\Lambda \cdot \ell_i(\boldsymbol{\theta}_t)$ and $\|\nabla \mathcal{L}(\boldsymbol{\theta}_t)\|^2 \leq 2\Lambda \cdot \mathcal{L}(\boldsymbol{\theta}_t)$

Therefore,

$$E_{\mathcal{S}_t}[\|\nabla \mathcal{L}(\boldsymbol{\theta}_t; \mathcal{S}_t)\|^2] \leq \frac{1}{b^2} \sum_{i=1}^{b} E_{\boldsymbol{x}_i}[2\Lambda \ell_i(\boldsymbol{\theta}_t)] + \frac{b-1}{b} 2\Lambda \mathcal{L}(\boldsymbol{\theta}_t) \tag{5}$$

$$= \frac{1}{b} 2\Lambda \mathcal{L}(\boldsymbol{\theta}_t) + \frac{b-1}{b} 2\Lambda \mathcal{L}(\boldsymbol{\theta}_t) \tag{6}$$

$$= 2\Lambda \cdot \mathcal{L}(\boldsymbol{\theta}_t) \tag{7}$$

$\square$

### A.5 PROOF OF THEOREM 1

*Proof.* Given AS 2 and Lemma 3, $\mathcal{L}(\boldsymbol{\theta}_t)$ can be bounded by $\mathcal{L}(\boldsymbol{\theta}_t) \leq \mathcal{L}(\boldsymbol{\theta}_{t-1}) + \nabla \mathcal{L}(\boldsymbol{\theta}_{t-1})^T (\boldsymbol{\theta}_t - \boldsymbol{\theta}_{t-1}) + \frac{\Lambda}{2} \|\boldsymbol{\theta}_t - \boldsymbol{\theta}_{t-1}\|^2$ for $\forall \boldsymbol{\theta}_{t-1}, \boldsymbol{\theta}_t$

In SLIM-QN, $\boldsymbol{\theta}_t = \boldsymbol{\theta}_{t-1} - \eta_{t-1} \hat{H}_k^{-1} \nabla \mathcal{L}(\boldsymbol{\theta}_{t-1}; \mathcal{S}_{t-1})$. Therefore, we can upper bound $\mathcal{L}(\boldsymbol{\theta}_t)$ as:

$$\mathcal{L}(\boldsymbol{\theta}_t) \leq \mathcal{L}(\boldsymbol{\theta}_{t-1}) - \eta_{t-1} \cdot \nabla \mathcal{L}(\boldsymbol{\theta}_{t-1})^T \hat{H}_k^{-1} \nabla \mathcal{L}(\boldsymbol{\theta}_{t-1}; \mathcal{S}_{t-1})$$
$$+ n_{t-1}^2 \frac{\Lambda}{2} \left\| \hat{H}_k^{-1} \nabla \mathcal{L}(\boldsymbol{\theta}_{t-1}; \mathcal{S}_{t-1}) \right\|^2$$
$$\leq \mathcal{L}(\boldsymbol{\theta}_{t-1}) - \eta_{t-1} \cdot \nabla \mathcal{L}(\boldsymbol{\theta}_{t-1})^T \hat{H}_k^{-1} \nabla \mathcal{L}(\boldsymbol{\theta}_{t-1}; \mathcal{S}_{t-1})$$
$$+ n_{t-1}^2 \frac{\Lambda \Xi^2}{2} \|\nabla \mathcal{L}(\boldsymbol{\theta}_{t-1}; \mathcal{S}_{t-1})\|^2$$

Since $\hat{H}_k^{-1}$ is independent with $\mathcal{S}_{t-1}$, we take expectation w.r.t $\mathcal{S}_{t-1}$ and $\mathcal{S}_t$,

$$E_{\mathcal{S}_t}[\mathcal{L}(\boldsymbol{\theta}_t)|\mathcal{S}_{t-1}] \leq \mathcal{L}(\boldsymbol{\theta}_{t-1}) - \eta_{t-1} \cdot \nabla \mathcal{L}(\boldsymbol{\theta}_{t-1})^T \hat{H}_k^{-1} E_{\mathcal{S}_{t-1}}[\nabla \mathcal{L}(\boldsymbol{\theta}_{t-1}; \mathcal{S}_{t-1})]$$
$$+ n_{t-1}^2 \frac{\Lambda \Xi^2}{2} E_{\mathcal{S}_{t-1}}[\|\nabla \mathcal{L}(\boldsymbol{\theta}_{t-1}; \mathcal{S}_{t-1})\|^2]$$
$$= \mathcal{L}(\boldsymbol{\theta}_{t-1}) - \eta_{t-1} \cdot \nabla \mathcal{L}(\boldsymbol{\theta}_{t-1})^T \hat{H}_k^{-1} \nabla \mathcal{L}(\boldsymbol{\theta}_{t-1})$$
$$+ n_{t-1}^2 \frac{\Lambda \Xi^2}{2} E_{\mathcal{S}_{t-1}}[\|\nabla \mathcal{L}(\boldsymbol{\theta}_{t-1}; \mathcal{S}_{t-1})\|^2]$$

According AS 1, we have

$$L(\boldsymbol{\theta}_{t-1}) \leq \tfrac{1}{\lambda} \|\nabla \mathcal{L}(\boldsymbol{\theta}_{t-1})\|^2$$

According to Lemma 4, we have

$$E_{\mathcal{S}_{t-1}}[\|\nabla \mathcal{L}(\boldsymbol{\theta}_{t-1}; \mathcal{S}_{t-1})\|^2] \leq 2\Lambda \mathcal{L}(\boldsymbol{\theta}_{t-1})$$

Therefore, $E_{\mathcal{S}_t}[\mathcal{L}(\boldsymbol{\theta}_t)|\mathcal{S}_{t-1}] \leq \mathcal{L}(\boldsymbol{\theta}_{t-1}) - \eta_{t-1} \lambda \xi \mathcal{L}(\boldsymbol{\theta}_{t-1}) + \eta_{t-1}^2 \Lambda^2 \Xi^2 \mathcal{L}(\boldsymbol{\theta}_{t-1})$.

After simply regrouping, we can get

$$E_{\mathcal{S}_t}[\mathcal{L}(\boldsymbol{\theta}_t)|\mathcal{S}_{t-1}] \leq (1 - \eta_{t-1} \lambda \xi + \eta_{t-1}^2 \Lambda^2 \Xi^2)[\mathcal{L}(\boldsymbol{\theta}_{t-1}]$$

Apply total expectation rule w.r.t $\mathcal{S}_t$, we have

$$E_{\mathcal{S}_t}[\mathcal{L}(\boldsymbol{\theta}_t)] \leq (1 - \eta_{t-1} \lambda \xi + \eta_{t-1}^2 \Lambda^2 \Xi^2) E_{\mathcal{S}_{t-1}}[\mathcal{L}(\boldsymbol{\theta}_{t-1})]$$

$\square$

## B    SETTINGS FOR MODEL TRAINING ON IMAGENET

### B.1    RESNET-50

We train ResNet-50 on ImageNet in a multi-GPU platform, which has 8 Nvidia Quadro RTX 5000 GPUs. PyTorch ($\geq$1.8) and the Distributed Data Parallel (DDP) communication package is used during training. Table 3 lists hyperparameters used in SGD, KFAC, and SLIM-QN. Initial learning rate is $0.1$, decaying by a factor of 10 at $30, 60, 90$th epoch. We use a learning warmup for in the first 2 epochs. Batch size is 256. In SLIM-QN, we use a smaller weight decay ($wd$) as SLIM-QN considers weight decay when conditioning gradients. Therefore a smaller weight decay can achieve as strong regularization as SGD with $wd = 0.0005$. For the Hessian update frequency ($L$), in SLIM-QN we update the Hessian for every 30 mini-batch iterations; while for KFAC, considering the compute cost, we update the Hessian for every 50 iterations. Lower and upper threshold for restraining eigenvalues in the Hessian ($\sigma_L, \sigma_H$) is set to be $(0.01, 1)$ in all experiments. Initial damping ($1 - \tau_0$) is $0.05$, then adapted during training according to equation 4. We run 5 random seeds for each optimizer.

Table 3: Hyperparameters for SGD, KFAC, SLIM-QN on ResNet-50/ImageNet

| Optimizer | $lr$ | momentum | $wd$ | damping | $\beta_1/\beta_2$ | $L$ | $M$ |
|---|---|---|---|---|---|---|---|
| SGD | 0.1 | 0.9 | 0.0005 | - | - | - | - |
| KFAC | 0.1 | 0.9 | 0.0002 | 0.001 | - | 50 | - |
| SLIM-QN | 0.1 | 0.9 | 0.0002 | 0.05 | 0.9/0.9 | 30 | 10 |

$\beta_1/\beta_2$: momentum for the Hessian; $L$: frequency for updating the Hessian; $M$: length of history vector $(\boldsymbol{s}, \boldsymbol{y})$

### B.2    VIT

We use a small Vision Transformer model with 6 layers, 8 attention heads, a patch size of 16, and both hidden and MLP dimension of $512$ for a total of about $10M$ parameters. We train for 90 epochs with linear learning rate warmup in the first 5 epochs, and decay the learning rate at $80$ epochs for SLIM-QN. For SGD, we train for 100 epochs, decaying the learning rate at $30, 60, 90$ epochs. A batch size of 1024 is used for both algorithms. We perform 3 runs with different random seeds. Table 4 shows the selected hyperparameters.

Table 4: Hyperparameters for SGD and SLIM-QN on ViT/ImageNet

| Optimizer | $lr$ | momentum | $wd$ | damping | $\beta_1/\beta_2$ | $L$ | $M$ |
|---|---|---|---|---|---|---|---|
| SGD | 0.1 | 0.9 | 0.0001 | - | - | - | - |
| SLIM-QN | 0.1 | 0.9 | 0.0 | 0.01 | 0.99/0.99 | 100 | 20 |

$\beta_1/\beta_2$: momentum for the Hessian; $L$: frequency for updating the Hessian; $M$: length of history vector $(\boldsymbol{s}, \boldsymbol{y})$

## C    EXPERIMENTS ON CIFAR-10

### C.1    RESNET-18

Hyperparameter is shown as in Table 5. Initial learning rate is $0.1$, decaying by a factor of 10 at 150th epoch. For both SGD and SLIM-QN we used a linear learning rate warmup for 5 epochs. Batch size is 256 for both SGD and SLIM-QN. Figure 5 shows convergence on ResNet-18 using SGD and SLIM-QN. On small dataset CIFAR-10, SGD and SLIM-QN both deliver fast convergence at early stages, while SLIM-QN is slightly better. Moreover, SLIM-QN achieves more faster convergence and high validation accuracy at later stages.

### C.2    VIT

We use the same ViT model as in the ImageNet experiment, but with a patch size of 16. Table 6 shows the hyperparameters used in the CIFAR-10 experiments. For both SGD and SLIM-QN we used a linear learning rate warmup for 5 epochs. We perform 3 runs with different random seeds.

Table 5: Hyperparameters for SGD, SLIM-QN on ResNet-18/CIFAR-10

| Optimizer | $lr$ | momentum | $wd$ | damping | $\beta_1/\beta_2$ | $L$ | $M$ |
|---|---|---|---|---|---|---|---|
| SGD | 0.1 | 0.9 | 0.0005 | - | - | - | - |
| SLIM-QN | 0.1 | 0.9 | 0.0005 | 0.05 | 0.9/0.9 | 100 | 10 |

$\beta_1/\beta_2$: momentum for the Hessian; $L$: frequency for updating the Hessian; $M$: length of history vector $(\boldsymbol{s}, \boldsymbol{y})$

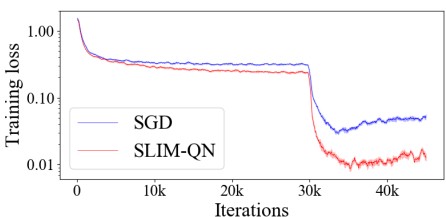 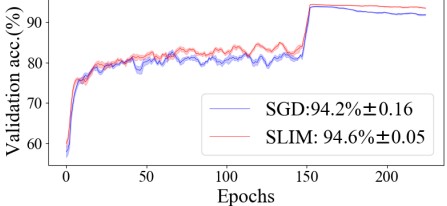

Figure 5: Convergence on ResNet-18/CIFAR-10 using SGD and SLIM-QN.

We use a batch size of 1024 and train for 90 epochs and decay the learning rate by a factor of 10 at 100 epochs for SLIM-QN. We train for 150 epochs and decay the learning rate at 140 epochs for SGD. Experimental results on CIFAR-10 using Vision Transformer are depicted in Fig. 6. We observe that SLIM-QN converges to a solution with better generalization than SGD and in overall less iterations. On small datasets such as CIFAR-10 we do not observe significant early-stage speedup on ViT, which is consistent with ResNet-18/CIFAR-10 experiments. Moreover, ViT is better suited for large-scale vision datasets due to the weaker implicit bias resulting from the non-convolutional architecture.

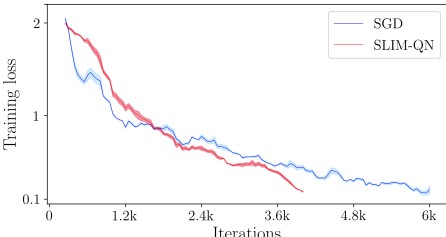 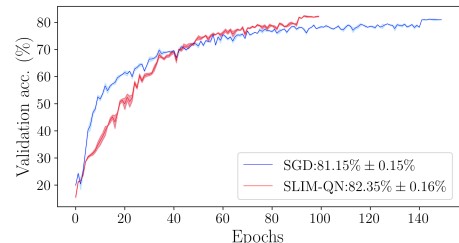

Figure 6: Convergence on ViT/CIFAR-10 using SGD and SLIM-QN.

Table 6: Hyperparameters for SGD, SLIM-QN on ViT/CIFAR-10 with batch size 1024

| Optimizer | $lr$ | momentum | $wd$ | damping | $\beta_1/\beta_2$ | $L$ | $M$ |
|---|---|---|---|---|---|---|---|
| SGD | 0.1 | 0.9 | 0.0001 | - | - | - | - |
| SLIM-QN | 0.025 | 0.9 | 0.0001 | 0.01 | 0.99/0.99 | 100 | 10 |

$\beta_1/\beta_2$: momentum for the Hessian; $L$: frequency for updating the Hessian; $M$: length of history vector $(\boldsymbol{s}, \boldsymbol{y})$

# D  STOCHASTIC TRAINING USING THE CLASSICAL L-BFGS

In this section, we demonstrate that the classical L-BFGS suffers convergence instability in stochastic training, even using large batch sizes. We train ResNet-18 on CIFAR-10, and vary batch size from 64 to 2048. Learning rate decays from 0.1 to 0.001 in 100 epochs. Weight decay is 0.0005. The Hessian approximation is updated using the L-BFGS formula for every 50 iterations, with at most 10 history vectors.

Fig. 7 shows training accuracy w.r.t iterations (log scale). We can observe that L-BFGS fails to converge under various batch sizes. Training using large batches though is a little more stable than small ones at the beginning, still diverges due to stochastic noises or sudden changes in loss landscape. Therefore, large batch only is not a sufficient solution to address convergence instability in the classical L-BFGS.

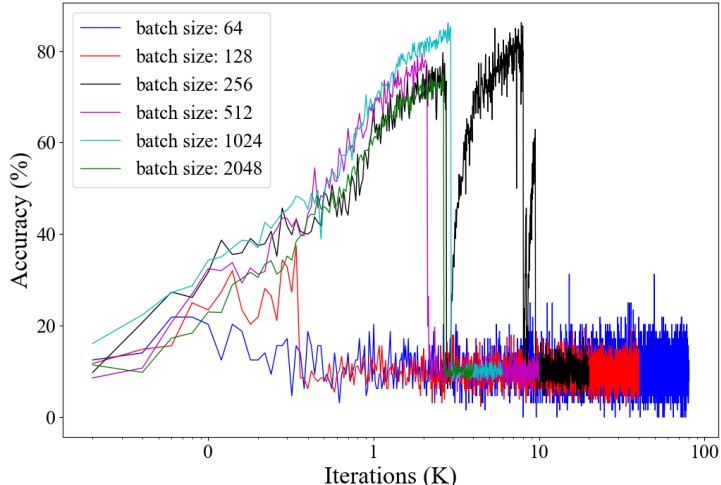

Figure 7: Convergence on ResNet-19/CIFAR-10 using the classical BFGS.

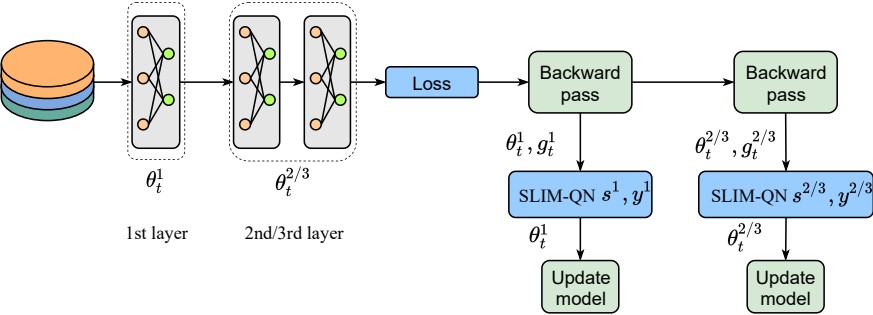

Figure 8: Block-wise SLIM-QN for distributed systems. Models are divided into blocks, which are then optimized by SLIM-QN in multiple nodes.

# E   SLIM-QN IN DISTRIBUTED SYSTEMS

As mentioned in Sec. 4.2, SLIM-QN requires $O(2M \|\boldsymbol{\theta}\|)$ storage to store required history statistics $\boldsymbol{s}_k$ and $\boldsymbol{y}_k$. When training models using data parallelism, each node further needs to store the whole copy of these history vectors. Such high memory footprints make it difficult to be used to train very large models such as BERT and GPT (Devlin et al., 2018; Radford et al., 2018). To address such a limitation, we propose a block-wise SLIM-QN which is much more memory-efficient in distributed systems.

As shown in Fig. 8, for a neural network, we divide model parameters into multiple blocks, where each block might consist of one or more layers. During training, these blocks are optimized in parallel using independent SLIM-QN optimizers. In a distributed system, each compute node can conduct optimization on one or more blocks, depending on its capabilities. Furthermore, the model can be divided in a way such that each node can store the required statistics $\boldsymbol{s}_k$ and $\boldsymbol{y}_k$ for at least one block. Combining with ZeRO data parallelism design (Rajbhandari et al., 2020), block-wise SLIM-QN are capable of training very large models as long as there are enough compute nodes.

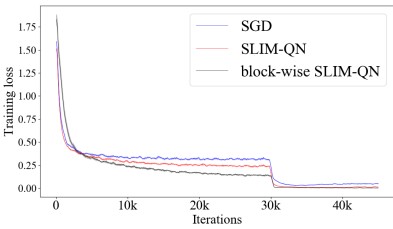 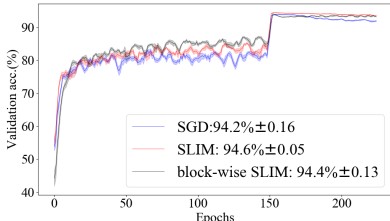

Figure 9: Convergence on ResNet-18/CIFAR-10 using SGD, SLIM-QN, and block-wise SLIM-QN

### E.1 CONVERGENCE GUARANTEES

In this section, we prove that block-wise SLIM-QN also converges in a linear rate given assumption in Sec. 4.1. Furthermore, the convergence property is guaranteed in any arbitrary way of dividing models.

**Theorem 2.** *Assume AS 1-3 hold at each iteration $t$ of block-wise SLIM-QN with mini-batch input $\mathcal{S}_t$, and the $k$-th Hessian update for block $i(i = 1, \cdots, p)$, $\hat{B}_k^i$ ($\hat{B} = \hat{H}^{-1}$) during the optimization is bounded by $\xi_i I \preceq \hat{B}_k^i \preceq \Xi_i I$, then the expectation of $\mathcal{L}(\boldsymbol{\theta}_t)$ satisfies*

$$E_{\mathcal{S}_t}[\mathcal{L}(\boldsymbol{\theta}_t)] \leq \alpha_{t-1} E_{\mathcal{S}_{t-1}}[\mathcal{L}(\boldsymbol{\theta}_{t-1})],$$

*where $\alpha_{t-1} = 1 - \eta_{t-1}\lambda\xi + \eta_{t-1}^2 \Lambda^2 \Xi^2$, $\xi = \min \xi_i$, $\Xi = \max \Xi_i$.*

*Proof.* The bound of $\hat{B}_k^i$ for block $i$ is guaranteed by SLIM-QN. Then according to the proof in Theorem 1, if we can prove $\hat{H}_k^{-1} = \text{diag}(\hat{B}_1, \hat{B}_2, \cdots, \hat{B}_p)$ is also bounded, then we can show block-wise SLIM-QN converges in a linear rate.

For any arbitrary vector $\boldsymbol{x} \neq \boldsymbol{0}$, $\boldsymbol{x}^T \cdot \hat{H}_k^{-1} \cdot \boldsymbol{x}$ can be written as

$$(\boldsymbol{x}^{1^T} \cdot \hat{B}_1, \cdots, \boldsymbol{x}^{p^T} \cdot \hat{B}_p) \cdot (\boldsymbol{x}^{1^T}, \cdots, \boldsymbol{x}^{p^T})^T = \sum_{i=1}^{p} \boldsymbol{x}^{i^T} \cdot \hat{B}_i \cdot \boldsymbol{x}^i,$$

where $\boldsymbol{x}^i$ is a sub-vector corresponding to block $i$.

Let $\xi = \min \xi_i$ and $\Xi = \max \Xi_i$, therefore, $\xi \leq \boldsymbol{x}^T \cdot \hat{H}_k^{-1} \cdot \boldsymbol{x} \leq \Xi$.

Following the same proof as in Theorem 1, block-wise SLIM-QN also converges in a linear rate. $\square$

### E.2 EMPIRICAL ANALYSES

We evaluate block-wise SLIM-QN on ResNet-18/CIFAR-10 using 5 random seeds. Layers in a ResBlock are grouped into one block. Initial learning rate is 0.1, decaying by a factor of 10 at 150th epoch. We used a linear learning rate warmup for 5 epochs. Batch size is 256 for all runs. Table 7 list detailed hyper parameters settings. As shown in Fig. 9, block-wise SLIM-QN achieves even slightly faster convergence performance as SLIM-QN. Both SLIM-QN and the block-wise version achieve higher validation accuracy compared to SGD.

Due to the limited time, we have not completed experiments on large models/datasets. We will add that in the future.

Table 7: Hyperparameters for SGD, SLIM-QN and block-wise SLIM-QN on ResNet-18/CIFAR-10

| Optimizer | $lr$ | momentum | $wd$ | damping | $\beta_1/\beta_2$ | $L$ | $M$ |
|---|---|---|---|---|---|---|---|
| block-wise SLIM-QN | 0.1 | 0.9 | 0.0003 | 0.01 | 0.9/0.9 | 50 | 10 |

$\beta_1/\beta_2$: momentum for the Hessian; $L$: frequency for updating the Hessian; $M$: length of history vector $(\boldsymbol{s}, \boldsymbol{y})$

## F   HYPERPARAMETER TUNING

SLIM-QN involves additional hyperparameters besides common parameters in SGD: learning rate, momentum, and weight decay. While tuning the hyperparameters is not a huge burden since some parameters are fixed in all the experiments, such as lower and upper threshold for restraining eigenvalues in the Hessian: $\sigma_L, \sigma_H$, and length of history vectors: $M$. For the rest of parameters: damping ($\tau_0$), momentum ($\beta_1, \beta_2$) and update frequency ($L$) for the Hessian, it is easy to conduct a grid search on a small models and datasets, explore how these parameters affect optimization, then apply them to large-scale model training. For example, after training on CIFAR-10, we found that small $L$ improves generalization performance, and large damping stabilizes optimization but causes optimizer to behave like SGD. With these notions, it is easy to tune these parameters, and then achieve optimal convergence performance and accuracy.

