# OpenReview forum: "SLIM-QN: A Stochastic, Light, Momentumized Quasi-Newton Optimizer for Deep Neural Networks"
_ICLR.cc/2022/Conference — ICLR 2022 Submitted_

### Official Review · Reviewer_doqb · 2021-10-24

**Correctness:** 3
**Technical Novelty And Significance:** 2
**Empirical Novelty And Significance:** 2
**Recommendation:** 3
**Confidence:** 4

**Main Review:**

The idea of combining momentum with L-BFGS has been explored before, e.g. in Chang et al "An Accelerated Linearly Convergent
Stochastic L-BFGS Algorithm", 2019. I would like to see a comparison between the proposed method and that algorithm, they seem quite different. Damping to avoid very small eigenvalues of the preconditioners is also a standard technique.

In table 2, the authors compare the time and memory of SGD, KFAC against SLIM-QN. The memory requirements of the proposed method are quite high, up to 20-40 times more than SGD, this would prevent this method from being used for the large models such as BERT that are commonplace in ML practice today (Resnet-50 has 23M parameters, whereas BERT has 340M). Is it possible to sketch the historical gradients and parameters to reduce this memory significantly?

The experiments compare the proposed method against KFAC and SGD only, ignoring the more commonly used optimizers such as Adam and LAMB, and completely ignoring other second order optimizers. The new method matches SGD on Resnet-50 in 95 epochs with a batchsize of 256, and is slightly more expensive, and neither achieved the ML-Perf benchmark accuracy of 75.9, which can now be achieved in less than 50 epochs. Also, these other implementations use much larger batch-sizes like 32K or 64K (larger batches usually means larger number of epochs for the same accuracy) --- thus 95 epochs seems far too high. I would like to see the results on larger batch-sizes anyway as I suspect that the benefits of momentum and damping may not be as significant.

**Summary Of The Paper:**

The paper improves L-BFGS by adding momentum to the gradients and past parameters, and adding a damping term to the Hessian to improve the condition number. This ensures that the Hessian inverse is stable (i.e. is not very much affected by individual mini-batches) and its eigenvalues are well controlled.

The authors prove the convergence of their algorithm, using standard techniques. They also provide space and time complexity estimates for their algorithm, and compare it against the estimates for SGD and KFAC.

The authors run their algorithm on Imagenet with Resnet-50, and compare its performance with SGD and KFAC. They also perform experiments to show that both damping and momentum are important for convergence: without momentum, the validation performance oscillates more, and without damping the algorithms diverges.

**Summary Of The Review:**

Overall I feel that both the novelty and impact of the paper are somewhat limited. The key techniques proposed have been studied before, and the resulting method is only practical for small problems. Further the gains from this method are rather small compared to current optimizers, and the authors do not compare against the best optimizers.

---

> ### Author Response · Authors · 2021-11-19
> **Response of the reviewer's comments**
>
> We welcome the reviewer's concerns about the novelty and downsides of SLIM-QN. We would like to take this opportunity to address the reviewer’s concerns.
>
> - The idea of combining momentum with L-BFGS has been explored before.
>
> It is true that [1] uses momentum in L-BFGS to stabilize the optimization. However, the momentum used in [1] is applied to parameter updates, rather than directly to parameter or gradient *changes* ($s_k$ and $y_k$) for deriving the Hessian inverse. It does not help reduce stochastic noise in the Hessian approximation, which we have shown in Sec 5.2 is very critical to convergence stability. In fact, in [1], to obtain the stable Hessian approximation, authors use a separate batch of inputs $b_H$ to compute parameter and gradient changes like other L-BFGS variants [2]. Combining with full-batch gradient, these additional operations significantly increase the total compute cost. These critical compute bottlenecks are indeed the motivation for us to design a lightweight second-order optimizer. To give a better overview of the differences between stochastic L-BFGS in [1] and SLIM-QN, we add a comparison in Table 2.
>
> - Memory requirements of SLIM-QN are quite high.
>
>  We appreciate the comments on the memory cost of SLIM-QN. As the reviewer stated, the memory cost of SLIM-QN is 20~40 times higher than SGD, which can be a main bottleneck when it is applied to large models, such as BERT.
>
> To resolve such an issue, we propose a distributed SLIM-QN design in the updated manuscript (Appendix E). We would like to sketch the high-level idea here: in the distributed design, a model is divided into several blocks, and each block is distributed to a compute node, and then optimized using SLIM-QN in parallel. As long as we have enough compute nodes, large models such as BERT can be divided into small blocks such that each compute node has enough memory to store one block and perform optimization. Therefore, the memory bottleneck can indeed be resolved. We also provide theoretical and empirical analysis on such a distributed design. Please check the details in the updated manuscript.
>
> - Compare against KFAC and SGD only, ignore more commonly used optimizers such as Adam and LAMB.
>
> As we explained in the paper, SGD is commonly used in image classification tasks. Other adaptive methods such as Adam are known to converge faster but likely loose the accuracy [3,4]. Hence we did not include Adam as one of the baselines. As for LAMB, it is designed for very large-batch training, which goes beyond the scope of our current work. However, as the reviewer suggested, we will consider extending SLIM-QN to very large-batch settings using our proposed distributed design.
>
> [1] Chang, D., Sun, S. and Zhang, C., An accelerated linearly convergent stochastic L-BFGS algorithm. IEEE transactions on neural networks and learning systems, 2019.
>
> [2] Moritz, P., Nishihara, R. and Jordan, M., A linearly-convergent stochastic L-BFGS algorithm. In Artificial Intelligence and Statistics, 2016.
>
> [3] You, Y., Li, J., Reddi, S., Hseu, J., Kumar, S., Bhojanapalli, S., Song, X., Demmel, J., Keutzer, K. and Hsieh, Large batch optimization for deep learning: Training bert in 76 minutes. arXiv preprint arXiv:1904.00962, 2019.
>
> [4] Defazio, A. and Jelassi, S.,  Adaptivity without compromise: a momentumized, adaptive, dual averaged gradient method for stochastic optimization. arXiv preprint arXiv:2101.11075, 2021.

---

### Official Review · Reviewer_bMXz · 2021-10-31

**Correctness:** 2
**Technical Novelty And Significance:** 2
**Empirical Novelty And Significance:** 3
**Recommendation:** 3
**Confidence:** 4

**Main Review:**

Strengths
- This work proposes useful techniques (momentum and adaptive damping) to stabilize the L-BFGS algorithm in training neural networks with a small mini-batch size (as I will describe below, the motivation of applying L-BFGS to small-batch training is not convincing though).
- This is the first empirical result showing that the second-order optimization method can achieve faster training time than SGD for training a Vision Transformer model.

Weaknesses
- The motivation of applying L-BFGS to small-batch training is not convincing.
    - The authors motivate using/stabilizing L-BFGS in small-batch training by stating that using a separate large batch to reduce stochastic noise “dramatically increases the computation cost and negates performance gains in wall-clock time”. But what if we train a model with a large batch size?
    - Zhang et al, 2019 [1] empirically showed that the “critical batch size”, in which the data efficiency (test accuracy vs number of examples a model consume) of mini-batch training decreases, of a second-order method (K-FAC) is larger than SGD and Adam. This means that we might be able to train a model by a second-order optimization with a large batch with the same number of epochs (i.e., the same number of total examples the model consume) as small-batch training with less stochastic noise.
    - Note that by accumulating multiple batches of mini-batch gradients before updating parameters, it is possible to train a model with arbitrary batch sizes even with a single processor. As long as the data efficiency is constant, the total computing cost (the total number of examples to calculate gradients) is also constant. So, simply using a large batch size may eliminate stochastic noise without additional computational and memory costs.
    - Therefore, to motivate small-batch training, it is necessary to present that L-BFGS has problems in large-batch training such as generalization performance degradation (as in large-batch SGD [2]).
- Parts of the discussion of the pros and cons of K-FAC and SLIM-QN (L-BFGS) are not appropriate.
    - In Figure 2, only the computational/memory overheads of K-FAC is highlighted by red color. However, the overheads of SLIM-QN (L-BFGS) for “Model update” is actually much larger than K-FAC for most of the case. For example, assume $d_i=\sqrt{||\theta_i||}$, the memory overhead of K-FAC in “Model update” will be $2||\theta||$ while that of SLIM-QN (L-BFGS) is $M$ times larger (actually this should be $2(M+1)$ because SLIM-QN needs to keep the momentum of the parameter difference and gradient difference). As $M=10-20$ is typically used, this is problematic especially when the model size is huge (e.g., 175B parameters of GPT-3 model [3]). Therefore, SLIM-QN (L-BFGS) is not suitable for training large models. On the other hand, the inverse matrix calculations in K-FAC can be distributed across multiple processors, thus reducing the computation time [4]. In addition to this, if K-FAC can be used with a large-batch size, the total number of parameter updates (i.e., number of inverse calculations) will be reduced. When combined with inverse frequency reduction (i.e., $1/L$), the computational overhead of K-FAC becomes even smaller. SLIM-QN (L-BFGS) can be more efficient than K-FAC if the target model is small enough to allow $M$ to be sufficiently large. However, the absence of these discussion in the current manuscript gives an unfair impression.
    - The authors state that “Unlike KFAC, SLIM-QN can easily incorporate weight decay (wd) by simply adding wd”. I assume that the authors mean L2 regularization by “weight decay” because we can easily incorporate weight decay to K-FAC as well (i.e., $g=g_{kfac}+\alpha\theta$). The previous works suggest that weight decay is better (better generalization performance, increased effective learning rate = training speed) than L2 regularization for Adam [5] and K-FAC [6]. Therefore, it is not a major problem for K-FAC that L2 regularization (the authors probably call it weight decay) is difficult to integrate.
    - The authors argue that K-FAC “need to run backward passes multiple times”. This is not the case when the empirical Fisher is used as an estimate of the Fisher information matrix in K-FAC. The empirical Fisher can be calculated during the backward pass of the empirical loss. Osawa et al. [4] showed that K-FAC with empirical Fisher can achieve the same accuracy with an estimate Fisher by one Monte-Carlo sampling from the model’s predictive distribution in training ResNet-50 on ImageNet-1k classification (the same thing does not necessarily happen in different settings, though).
- SLIM-QN’s practical usefulness is still questionable.
    - The authors state that “it is easy to conduct a grid search on a small models and datasets, explore how these parameters affect optimization, then apply them to large-scale model training”. I believe that the authors need to quantitatively evaluate how “easy” this process is. The behavior of the Hessian during training may change significantly when the model size or data size changes.

Other comments
- The authors introduce their method as “SLIM-QN directly approximates the Hessian inverse using past parameters and gradients, without explicitly constructing the Hessian matrix and then computing its inverse”. However, this is just a description of BFGS or L-BFGS. I think it would be easier for readers to understand the differences of SLIM-QN from L-BFGS (the contribution of this work) if the explanation is something like “SLIM-QN is L-BFGS with additional techniques such as momentum and adaptive damping.” In fact, SLIM-QN is implemented as the “LBFGSOptimizer” in the author’s code.

References
- [1] G. Zhang et al. Which Algorithmic Choices Matter at Which Batch Sizes? Insights From a Noisy Quadratic Model. https://arxiv.org/abs/1907.04164, 2019.
- [2] N. S. Keskar et al. On Large-Batch Training for Deep Learning: Generalization Gap and Sharp Minima. https://arxiv.org/abs/1609.04836, 2017.
- [3] T. B. Brown et al. Language Models are Few-Shot Learners. https://arxiv.org/abs/2005.14165, 2020.
- [4] K. Osawa et al. Scalable and Practical Natural Gradient for Large-Scale Deep Learning. https://arxiv.org/abs/2002.06015, 2020.
- [5] I. Loshchilov and F. Hutter. Decoupled Weight Decay Regularization. https://arxiv.org/abs/1711.05101, 2019.
- [6] G. Zhang et al. Three Mechanisms of Weight Decay Regularization. https://arxiv.org/abs/1810.12281, 2018.


**Summary Of The Paper:**

This work proposes techniques that stabilize the small-batch training of neural networks by L-BFGS optimizer. To reduce the stochastic noise of gradient and Hessian, they suggest taking the momentum of the parameter difference and gradient difference used to estimate the inverse-Hessian-gradient product in L-BFGS. In training Resnet models and Vision Transformer models on image classification tasks (CIFAR-10, ImageNet-1k) with small batch size, the proposed SLIM-QN method (L-BFGS + momentum + adaptive damping) achieves a faster training time than SGD.

**Summary Of The Review:**

This work provides interesting empirical results that a second-order optimization method (other than K-FAC) can achieve fast training in the ImageNet scale. Also, this is the first result that a Vision Transformer model is trained by a second-order optimization method faster than SGD. These results can be a data point for future studies on more efficient optimization methods for deep neural networks. However, the motivation for small-batch training, which is the main target of this study, is not convincing. Moreover, the validity of the comparison between the proposed and existing methods (K-FAC) is questionable.

---

> ### Author Response · Authors · 2021-11-19
> **Responses to the reviewer's comments**
>
> We thank the reviewer's insightful comments. Comments such as large batch on single processor, memory cost of SLIM-QN indeed helped improve the paper. However, we also think it is necessary to make clarifications that might help clarify some points for the reviewer.
>
> - The motivation of applying L-BFGS to small-batch training is not convincing.
>
> The discussion on batch size is not to motivate small-batch training. In fact, we try to analyze compute costs and locate main bottlenecks in L-BFGS variants. Estimating the Hessian with a separate large batch is used in [1,2], which leads to statements that L-BFGS variants “drastically increase the compute cost and negate performance gains in wall-clock time”.
>
> As for the question of “what if we train a model with a large batch size”, we did experiments when initiating lightweight QN. We train ResNet-18 on CIFAR10 with various batch sizes. The classical L-BFGS with large batches is easy to diverge (see appendix D). Hence, we believe that large batch size only is not a sufficient solution to address convergence instability in L-BFGS. This is similar to first-order methods, where momentum is still used in large batches.
>
> - Discussions of the pros and cons of KFAC and SLIM-QN are not appropriate.
>
>     * We acknowledge that the highlights of memory costs in Tables 2 might not be appropriate. In the case the reviewer pointed out, memory costs of SLIM-QN are higher than KFAC. However, we would like to discuss further and argue that the memory cost of SLIM-QN is not “much higher than KFAC” for common scenarios. \
> In CNNs, for a convolution layer with $c_o * c_i * k * k$ kernels ($c_o$/$c_i$: output/input channels), KFAC decomposes kernels into $c_o * c_o$, $c_i k^2 * c_i k^2$ sub matrices. Therefore, the memory cost of KFAC is $O(2(c_i^2k^4 +c_o^2))$, which is related to kernel sizes. SLIM-QN requires memory of $O(2Mc_o c_i k^2)$. Assume $c_i=c_o$ and $M=10$, memory costs of KFAC are higher when $k>3$. \
> In Transformer architectures, for an attention layer with hidden/head dimension of $d$,$d_h = d/h$ ($k$: number of heads), KFAC decomposes params into $d*d$ and $d_h * d_h$ sub-matrices. Therefore, KFAC consumes memory of $O(2(d_h^2 + d^2))$, while SLIM-QN needs $O(2Md d_h)$. Assuming $k=16$ and $M=10$, the memory cost of KFAC is still larger than SLIM-QN. \
> Therefore, the question of which one is more memory-intensive depends on models. We also have made necessary changes in the paper.
>
>     * For very large models such as GPT-3, SLIM-QN might not be suitable. To resolve this, we also propose a distributed solution (Appendix E). In the distributed design, we divide the model into several blocks, where each block is distributed to a single compute node and then optimized using SLIM-QN in parallel. Therefore,  the memory cost of SLIM-QN is resolved given sufficient number of compute nodes. We also provide theoretical and empirical analysis. Please see our updated manuscript.
>
>     * We thank the reviewer for providing references on L2 regularization and weight decay. We have changed to L2 regularization. \
> In the paper, we want to show that SLIM-QN can be flexibly applied with various regularizers, such as L2, and gradient regularization [3]. While KFAC is based on cross-entropy loss, which might not be able to directly include such regularizers.
>
>     * Based on Osawa's work, the reviewer’s argument is valid. We also would like to point out that there are works showing empirical Fisher is not a good approximation to the real Fisher [4]. Furthermore, Algorithm 2 in the original KFAC work [5] also suggested running multiple backwards to better estimate the required statistics. Therefore, we believe that one single empirical case might not justify that empirical Fisher is sufficient in all settings.
>
> - SLIM-QN’s practical usefulness is still questionable.
>
> The reviewer is correct that different models/datasets might present a different Hessian behavior. It is not wise to apply the same optimal hyper params. However, We can explore common properties when using SLIM-QN in small and large models. For example, more frequent Hessian updates with less history vectors leads to better generalization for ResNet18/CIFAR10, while such observation also holds on ResNet50/ImageNet.\
> Therefore, by knowing how these hyper parameters affect optimization in a general sense, we can have a more clear understanding of how to tune them toward a better performance.
>
> [1] Moritz, P. et al. A linearly-convergent stochastic L-BFGS algorithm. AISTATS, 2016
>
> [2] Chang, D. et al. An accelerated linearly convergent stochastic L-BFGS algorithm. TNNLS, 2019
>
> [3] Smith, S.L. et al. On the origin of implicit regularization in stochastic gradient descent. ICLR, 2021
>
> [4] Kunstner, F. et al. Limitations of the empirical fisher approximation for natural gradient descent. NeurIPS, 2019
>
> [5] Martens, J. et al. Optimizing neural networks with kronecker-factored approximate curvature. ICML, 2015

---

> > ### Comment · Reviewer_bMXz · 2021-11-21
> > **The novelty and efficacy of SLIM-QN are still questionable.**
> >
> >
> > Thanks for the responses and clarifications. However, the authors have not addressed my concerns, and the novelty and efficacy of SLIM-QN are still questionable.
> >
> > > Estimating the Hessian with a separate large batch is used in [1,2], which leads to statements that L-BFGS variants “drastically increase the compute cost and negate performance gains in wall-clock time.”
> >
> > Yes, I agree with this statement. But I was pointing out that the computing cost for L-BFGS has nothing to do with the batch size as long as the data efficiency is constant, and that’s why I questioned, ”what if we train a model with a large batch size?”.
> >
> > Thanks for conducting additional experiments with various batch sizes. But I am not convinced that “The classical L-BFGS with large batches is easy to diverge.” In Figure 7, the convergence of L-BFGS gets more stable when larger batch sizes are used. Hence, the authors’ statement that L-BFGS with a large batch size (up to 2048) “still diverges due to stochastic noises or sudden changes in loss landscape” is questionable. The training curves with large batch sizes (256-2048) are similar to “with only momentum” in Figure 4. Therefore, I wonder if the cause of divergence is improper damping value, and the stochastic noise is no longer a problem. If this is the case, it suggests that we can stabilize L-BFGS by simply using a large batch size and an appropriate damping value. As pointed out by Reviewer doqb, damping is a standard technique.
> >
> > Thanks for showing a comparison of memory costs of K-FAC and SLIM-QN.
> >
> > > the question of which one is more memory-intensive depends on models.
> >
> > Yes, so the memory overhead of SLIM-QN can be much more significant than K-FAC for a fully-connected layer with $d_i=\sqrt{||\theta_i||}$. On the other hand, in the attention and convolutional layers, the memory cost of K-FAC can be higher (if $M$ is small enough), as the authors showed. To clarify these relationships, comparisons of actual memory consumption for several neural network architectures (with various $M$) are helpful.
> >
> > >  we also propose a distributed solution (Appendix E).
> >
> > It seems that the authors now propose a model-parallel version of SLIM-QN with block-diagonal Hessian. It is unclear how practical this approach is. Even if we have $M$ GPUs, the memory cost per GPU is still $2||\theta||$, which is still too large for models such as BERT and GPT.
> >
> > > While KFAC is based on cross-entropy loss, which might not be able to directly include such regularizers (L2, and gradient regularization).
> >
> > K-FAC is not based on cross-entropy loss. K-FAC is used to train deep autoencoders in Martens and Grosse, 2015. As I pointed out, weight decay, which is easy to incorporate with any optimizers, is known to be more suitable than L2 regularization for K-FAC.
> >
> > > we believe that one single empirical case might not justify that empirical Fisher is sufficient in all settings.
> >
> > I agree with this statement. But at the same time, we could say that multiple back propagations are not necessarily required for K-FAC in all settings.
> >
> > > by knowing how these hyper parameters affect optimization in a general sense, we can have a more clear understanding of how to tune them toward a better performance.
> >
> >  I still believe that the authors need to quantitatively evaluate how “easy” this process is.

---

> > > ### Author Response · Authors · 2021-11-22
> > > **Further clarifications**
> > >
> > > Thanks for the quick responses. We would like to make some further clarification regarding 1) L-BFGS with large batch size; 2) distributed L-BFGS.
> > >
> > > - "In Figure 7, the convergence of L-BFGS gets more stable when larger batch sizes are used.The training curves with large batch sizes (256-2048) are similar to “with only momentum” in Figure 4"
> > >
> > > Note that the X-axis in Figure 7 is in log scale. Even though L-BFGS with large batch sizes are a little more stable, they still diverge at very early stages (~2k iterations). While L-BFGS with only momentum is much more stable before the learning rate decays (30k iterations). Such a comparison indicates that momentum is a more efficient way to stabilize optimization.
> > >
> > > - "If this is the case, it suggests that we can stabilize L-BFGS by simply using a large batch size and an appropriate damping value."
> > >
> > > As we shown in Figure 4, L-BFGS with damping (batch size is 256), though effectively preventing divergence, still cannot avoid notable fluctuations in training losa and test accuracy. With momentum added (blue curve in Figure 4), such fluctuations are significantly improved. We agree that L-BFGS might be more stable given sufficiently large batch sizes. An extreme case is to use full-batch gradients. However, we want to emphasize that momentum can be a more effective method to achieve the same goal.
> > >
> > > - "It seems that the authors now propose a model-parallel version of SLIM-QN with block-diagonal Hessian. It is unclear how practical this approach is."
> > >
> > > The reviewer is correct. The proposed distributed SLIM-QN is to approximate the block-diagonal Hessian, where each block might contain one or multiple layers. However, it is not a model-parallel design. Following a well-known data-parallel framework called DeepSpeed with ZeRO optimization [1,2] (ZeRO is used to train very large models such as Megatron-Turing [2]), we can easily implement block-wise SLIM-QN in a distributed system. Since each block is optimized separately, the required statistics ($s_k,y_k$) for each block are distributed to separate nodes. Given k blocks, the memory requirement for optimizer on each node is reduce to $\frac{2M}{k} \left\| \theta \right\|$. $k$ can be large enough so that the required statistics can be stored in one single GPU.
> > > This is similar to how adaptive methods such as Adam are implemented in DeepSpeed with ZeRO optimization. The only difference is that Adam only needs to store first- and second-order momentum, while block-wise SLIM-QN needs to store $s_k,y_k$.
> > >
> > > [1] Samyam Rajbhandari et al. ZeRO: memory optimizations toward training trillion parameter models. In Proceedings of the International Conference for High Performance Computing, Networking, Storage and Analysis (2019).
> > >
> > > [2] DeepSpeed. https://www.deepspeed.ai/

---

> > > > ### Comment · Reviewer_bMXz · 2021-11-22
> > > > **I will keep my assessment unchanged.**
> > > >
> > > > Thanks for the clarifications.
> > > >
> > > > > momentum can be a more effective method to achieve the same goal.
> > > >
> > > > Yes, it “can” be. But the authors need to quantitatively show how “effective” the proposed momentum approach is and how “easy” it is to tune hyperparameters compared to large-batch training or a naive momentum method (e.g., taking a momentum of preconditioned $g_t$ in Algorithm1).
> > > >
> > > >
> > > > ZeRO can indeed reduce the per-GPU memory cost of SLIM-QN (L-BFGS). However, SLIM-QN requires about $2M$ times more GPUs (or SLIM-QN can only train a $2M$ times smaller model on the same number of GPUs) than momentum SGD. Can investing $2M$ times more GPUs for SLIM-QN achieve faster training than investing $2M$ times more GPUs for momentum SGD or Adam training?

---

### Official Review · Reviewer_EjsS · 2021-11-01

**Correctness:** 4
**Technical Novelty And Significance:** 3
**Empirical Novelty And Significance:** 3
**Recommendation:** 6
**Confidence:** 3

**Main Review:**

As it for me,  it seems to be a nice work in developing second order method for large scale machine learning.  And I go over the proof quickly, it also seems ok.

**Summary Of The Paper:**


In this paper SLIM-QN  is proposed to  addresses two key barriers in existing second-order methods for large-scale DNNs: 1) the high computational cost of obtaining the Hessian matrix and its inverse in every iteration (e.g. KFAC); 2) convergence instability due to stochastic training (e.g. L-BFGS).
Converegence   results are provided in a stochastic setting. Numerical evaluations on real data  shown the advantage of in terms of  speed and accuracy.

**Summary Of The Review:**

In this paper SLIM-QN  is proposed. Converegence   results are provided in a stochastic setting. Numerical evaluations on real data  shown the advantage of in terms of  speed and accuracy.  I not quite sure why SLIM is faster than sgd (as reported  the wall-clock time) ? What is the stopping rule the authors used  for SGD ？

---

> ### Author Response · Authors · 2021-11-18
> **Responses to the reviewer's comments**
>
> We are happy to see that the reviewer recognizes the advantages of SLIM-QN over first-order methods such as SGD.
>
> For the reviewer’s concerns about the convergence performance of SLIM-QN in wall-clock time, we would like to take this chance and make further clarifications. SLIM-QN is a quasi-Newton (QN) method, which like other QN methods uses curvature (approximated) information to better optimize the loss function. Therefore it achieves faster per-iteration convergence compared to SGD. On the other hand, unlike other QN methods used in neural network (NN) optimization, SLIM-QN adds manageable compute costs when obtaining the approximated Hessian. Therefore, when training large-scale NN models, the total compute costs are just slightly higher than SGD (see Table 2). Such cost reduction in the Hessian approximation (together with the per-iteration convergence promises) naturally leads to better wall-clock convergence performance compared to SGD.

---

### Official Review · Reviewer_TATQ · 2021-11-02

**Correctness:** 4
**Technical Novelty And Significance:** 2
**Empirical Novelty And Significance:** 2
**Recommendation:** 5
**Confidence:** 3

**Main Review:**


### Strengths
1. The paper is well-written and easy to follow.
2. The proposed SLIM-QN has some computational and memory advantages.


### Concerns
1. In section 3.1, not sure why to consider $\mathcal{L} = \frac{1}{2}\left\lVert \theta \right\rVert^2$. It seems useless in practice.
2. AS1 and AS3 are very strong assumptions.
3. SLIM-QN is a method based on L-BFGS but Table2 and experiments lask the comparisons with the same kind of methods, e.g. BFGS and L-BFGS.

**Summary Of The Paper:**

This paper proposes SLIM-QN, a light stochastic quasi-Newton optimizer for training large-scale deep neural networks (DNNs). Based on previous results, SLIM-QN proposes to introduce momentum in Hessian updates to stabilize the training and adaptive damping mechanism to guarantee that the approximated Hessian is always positive definite. They prove the convergence of SLIM-QN under some assumptions. Some experimental results are also given.

**Summary Of The Review:**

Both the theoretical and experimental results are not sufficient. The assumptions of theoretical results are too strong and experimental results lack the comparison with the same kind of methods.

---

> ### Author Response · Authors · 2021-11-18
> **Responses to the reviewer's comments**
>
> We thank the reviewer for recognizing SLIM-QN’s main contributions in reducing computation and memory cost in current second-order optimization algorithms. Below are our response for each of the comments:
>
> 1. In section 3.1, the example of simple quadratic function seems useless in practice.
>
> The reviewer is correct that no such simple loss function exists in practice. However, we believe such a simple example is still needed to demonstrate differences among various quasi-Newton (QN) methods. Considering compute cost in practical problems, it is very difficult to show optimization trajectory (as in Figure 1) for some QN methods. Methods such as the exact Newton method are even impossible to implement in practical problems.  Therefore, a simple quadratic objective function, together with a simulated stochastic noise is a suitable example to provide some insight into how these QN methods behave in stochastic settings.
>
> 2. AS1 and AS3 are very strong assumptions.
>
> We understand the reviewer’s concern about the assumptions. We would like to make further clarification.
>
> For AS1, as we explained in Remark 2 after Theorem 1, this assumption is even weaker than the assumption of strong convexity. In strongly convex settings, we can easily derive that the norm of gradients is lower bounded by loss for a $\lambda > 0$, but not the other way around ( AS1). In addition, AS1 is also used in proving convergence in non-convex settings [1].
>
> For AS3, we acknowledge that it is uncommon. However, it is also reasonable when proving convergence in neural networks (NNs), especially for over-parameterized models. As stated in [2,3], current NN models are very large so that they usually exceed the size of the dataset. In such over-parameterized systems, it has been observed that most or all local minima are also global. Therefore, considering the practical and large-scale NNs we are dealing with, adopting such an assumption is still reasonable.
>
> 3. Lack comparison with the same kinds of methods.
>
> We thank the reviewer for the comparison of compute and memory cost. First, we would like to explain why we did not choose BFGS or L-BFGS as the baselines. When choosing baseline methods, our thinking is that we need to choose methods that are applicable to NN optimizers. However, the classical BFGS/L-BFGS methods cannot be directly applied to NN problems as they suffer convergence instability in stochastic settings (See ablation study in Sec 5.2 and Appendix D). Therefore, they are not considered as the baselines.
>
> That said, there are indeed several L-BFGS variants in NN optimizers, such as stochastic L-BFGS in [4, 5]. Compared to these variants, SLIM-QN uses much more light operations when obtaining the approximated Hessian, which is also our main contribution in the work. We will add such a comparison in the updated manuscript (in Table 2).
>
> [1] Kim, S. et al. Convergence of the Inexact Online Gradient and Proximal-Gradient Under the PL Condition. arXiv preprint arXiv:2108.03285, 2021.
>
> [2] Ma, S. et al. The power of interpolation: Understanding the effectiveness of SGD in modern over-parameterized learning. In the International Conference on Machine Learning,  2018.
>
> [3] Vaswani, S. et al. Fast and faster convergence of sgd for over-parameterized models and an accelerated perceptron. In The 22nd International Conference on Artificial Intelligence and Statistics 2019.
>
> [4] Moritz, P. et al. A linearly-convergent stochastic L-BFGS algorithm. In Artificial Intelligence and Statistics, 2016.
>
> [5] Chang, D.et al. An accelerated linearly convergent stochastic L-BFGS algorithm. IEEE transactions on neural networks and learning systems, 2019.

---

> > ### Comment · Reviewer_TATQ · 2021-11-22
> > **Responses to the authors and some other questions**
> >
> > Thanks for the responses and clarifications. Some further questions:
> >
> > - I wonder if you can prove the convergence under the convex and non-convex cases.
> > - In the proof of theorem 1, it is said "Given $L(\theta)$ is convex". I think you assume the PL condition instead of convexity?
> > - The notation of proof of Lemma 4 seems to have many problems. I do not know what is $m$ and $n$. The proof seems questionable to me.

---

> > > ### Author Response · Authors · 2021-11-22
> > > **Further clarification on assumptions and the convergence theorem.**
> > >
> > > - “I wonder if you can prove the convergence under the convex and non-convex cases.”
> > >
> > > The convergence analysis in Theorem 1 is for non-convex cases (under the assumption of PL condition). However, as we explained in Remark 2 right after Theorem 1, the theorem is also applied to strongly convex settings, since strong convexity implies $\left\| \nabla L(\theta) \right\|^2 \geq \lambda L(\theta)$ with  $\lambda > 0$.
> > >
> > > - Thanks for pointing out the misdescription, we have modified the section.
> > >
> > > - The reviewer is correct. There are some typos in Lemma 4. Both $m$ and $n$ means batch size $b$. We have changed them to $b$.

---

> > > > ### Comment · Reviewer_TATQ · 2021-11-22
> > > > **Further questions**
> > > >
> > > > - In the proof of Lemma 4, it seems you also assume data are independent, which is not clearly stated in the paper.
> > > > - It is not clear to me how you simplify the second part to $\frac{b-1}{b}\|| \nabla \mathcal{L}(\theta_t) \||^2$. Could you elaborate more?
> > > > - Besides, there is also a clear typo in the first line of proof of Lemma 4

---

> > > > > ### Author Response · Authors · 2021-11-22
> > > > > **Fix some unclear statement and typos in the paper**
> > > > >
> > > > > We appreciate your help in improving several descriptions.
> > > > >
> > > > > - The reviewer is correct, we have added necessary changes in Lemma 4 and Theorem 1.
> > > > >
> > > > > - We have added one more step (in red) and fixed some typos.
> > > > >
> > > > > - Thanks for pointing this out. We have corrected it.

---

> > > > > > ### Comment · Reviewer_TATQ · 2021-11-23
> > > > > > **Question about the proof**
> > > > > >
> > > > > > Hi,
> > > > > >
> > > > > > In the proof of Lemma 4, why is $E_{x_i, x_j}[\langle \nabla \ell_i(\theta_t), \nabla \ell_j(\theta_t) \rangle] = ||\nabla \mathcal{L}(\theta_t) ||^2$?
> > > > > >
> > > > > > This seems uncommon. Could you give more justification? It looks like you are putting the expectation into the inner product?

---

> > > > > > > ### Author Response · Authors · 2021-11-23
> > > > > > > **Further clarification**
> > > > > > >
> > > > > > > To simplify the notation, suppose $z_i=\nabla \ell_i(\theta_t, x_i)$, $z_j=\nabla \ell_j(\theta_t,x_j)$, and $z=\nabla\mathcal{L}(\theta_t)$.
> > > > > > >
> > > > > > > Then, the above equation is reduced as $E_{x_i, x_j} [\left \langle z_i, z_j \right \rangle] = E_{x_i,x_j}[\sum_{k=1}^n z_i(k)z_j(k)] = \sum_{k=1}^n E_{x_i,x_j}[z_i(k)z_j(k)]$, where $n$ is the dimension of the vector $z$.
> > > > > > >
> > > > > > > Since $x_i$, $x_j$ are independent, the expectation is further reduced to $\sum_{k=1}^n E_{x_i}[z_i(k)]E_{x_j}[z_j(k)] = \sum_{k=1}^n z(k)^2 = \left \| \nabla \mathcal{L}(\theta_t) \right \|^2 $.
> > > > > > >
> > > > > > > Please let us know if you have more questions.

---

### Official Review · Reviewer_aBmG · 2021-11-04

**Correctness:** 3
**Technical Novelty And Significance:** 2
**Empirical Novelty And Significance:** 3
**Recommendation:** 3
**Confidence:** 4

**Main Review:**

The paper is easy to follow and addresses an important problem. However, I have some major concerns about the assumptions and theoretical results.

1- The following text after equation (2) is a bit misleading. Specifically, the authors state that "By carefully choosing $s_k$, $\hat{H}_k^{-1}$ will converge to the real Hessian inverse at a linear rate for any strongly convex function (Rodomanov & Nesterov, 2021)." In (Rodomanov & Nesterov, 2021) the authors use a Greedy-based BFGS method and the nature of greedy BFGS allows the iterates to converge to the true Hessian. However, for any standard (classic) quasi-Newton method (including BFGS and L-BFGS) there is no conclusion that the Hessian inverse approximation converges to the exact Hessian inverse. All we can show for these algorithms is that the descent direction approaches the descent direction of Newton's method, i.e., $\hat{H}_k^{-1}\hat{g}_k\to {H}_k^{-1}{g}_k$. Since SLIM-QN is an L-BFGS-based quasi-Newton method we can’t expect that $\hat{H}_k^{-1}$ converges to ${H}_k^{-1}$. This point needs to be highlighted.

2- Assumption 3 is quite uncommon and strong for stochastic optimization analysis. In this assumption, the authors require the minimizer of each stochastic function to be the minimizer of the expected loss. This is indeed very strong, as it means that any stochastic descent direction (computed by a subset of samples) would move the iterates to the right minimum. Often in stochastic optimization theory we assume that in expectation we have a descent direction, but the required assumption implies that without taking expectation stochastic gradient or stochastic quasi-Newton directions will move the iterates towards the optimal solution.

3- The theoretical results, which are under some strong assumptions, do not showcase any improvement with respect to SGD or other quasi-Newton methods. To be more precise, under the considered assumption, one can easily establish a linear convergence rate for SGD or any stochastic quasi-Newton (SQN) method. The main unanswered question is whether SLIM-QN can provably outperform other stochastic methods (at least in a local neighborhood of the optimal solution) or not? Without such comparison or discussion the theoretical results are not informative and more importantly they don't improve the known bounds for SQN methods.

4- The authors suggest that the proposed SLIM-QN method leads to a more stable QN method. Intuitively, this is a valid point as they use the idea of momentum in the update of their QN method. However, there is no rigorous study (at least for a simple quadratic problem) to show that the Hessian approximation matrices of the proposed method are more stable compared to standard SQN methods.

5- The momentum and damping techniques are used to stabilize the noise created in the stochastic setting and they don't lead to reduction of computational cost or memory. The computational complexity is the use of L-BFGS method from the previous work as stated in Algorithm 1. The authors should highlight that the use of L-BFGS idea allows them to reduce the computational cost, and the momentum and damping techniques have nothing to do with the computational cost reduction. Their impact is only on stabilizing the learning process.

**Summary Of The Paper:**

In this paper the authors propose the SLIM-QN algorithm, a stochastic, momentum-based quasi-Newton method to train deep neural networks. SLIM-QN tries to address two shortcomings of second-order methods. The first barrier is the high computational cost of calculating the Hessian matrix and its inverse. SLIM-QN tackles this by directly approximating the Hessian inverse using past parameters and gradients without explicitly constructing the Hessian and computing its inverse. The second challenge is the convergence instability caused by the stochastic training. SLIM-QN overcomes this issue by using momentum in the Hessian updates as well as an adaptive damping mechanism. The authors provide theoretical results on the convergence of SLIM-QN for stochastic convex optimization problems, for the case that the objective function is smooth and satisfies the PL condition. They further conduct numerical experiments on large dataset to show the effectiveness of SLIM-QN.

**Summary Of The Review:**

Overall, I think the required assumptions are strong and the theoretical results are not significant.

---

> ### Author Response · Authors · 2021-11-18
> **Responses to the reviewer's comments**
>
> We appreciate the reviewer’s comments and suggestions on theoretical aspects in this paper. To begin with, we would like to re-emphasize our motivations for designing SLIM-QN. Current QN methods face many challenges in training large-scale neural networks, especially the compute/memory costs. For example, a well-known optimizer, KFAC, though enjoys faster per-iteration convergence compared to SGD, it is not necessarily fast in wall-clock time due to the fact that it needs additional efforts to obtain the curvature information and perform conditioning. L-BFGS suffers convergence instability when being naively used in training NNs (see ablation study in Sec 5.2 and Appendix D). To resolve such an issue in L-BFGS, people usually use full-batch gradients to reduce stochastic noise and a separate large batch to compute the Hessian [1,2].  These two additional operations add significant costs.
>
> In a nutshell, these additional operations that help achieve per-iteration convergence promises, unfortunately negate performance in wall-clock time.
>
> Given these issues, we design SLIM-QN. On one hand, we want SLIM-QN to achieve fast convergence as other QN methods; on the other hand, we refrain from adding costly operations as mentioned above. Therefore, the benefits of per-iteration convergence is also reflected in wall-clock time, which has not been seen in current QN methods.
>
> **Below is our detailed responses:**
>
> - Comments on convergence of BFGS/L-BFGS to the exact Hessian.
>
> We apologize for the misunderstanding. The reviewer is correct, classic BFGS/L-BFGS cannot guarantee convergence to the exact Hessian, which is also applied to SLIM-QN. We now have highlighted this point and made necessary changes in the manuscript (in red).
>
> - Assumption 3 is very strong.
>
> We acknowledge the reviewer’s concern: assumption 3 is uncommon in optimization analysis. However, this assumption holds in modern over-parameterized NNs and is observed in over-parameterized NNs [3, 4, 5]. As stated in [3], current SOTA NNs are very large so that they usually exceed the size of the dataset. In such over-parameterized NNs, it is observed that most or all local minima are also global and/or this assumption holds locally. Therefore, considering the practical NNs we are dealing with, adopting such an assumption is reasonable.
>
> - SLIM-QN does not showcase any convergence improvement w.r.t. SGD or other QN methods, even under AS 3.
>
> As we mentioned earlier, our goal is not to improve ‘the known bound of SQN methods’. Instead, we are trying to design a practical QN method with light compute/memory cost, especially when applied to large-scale NNs but still enjoys convergence guarantees on par with other QN methods. The aim of the theoretical guarantees is to show the algorithm is convergent under reasonable assumptions. So in that sense we agree that SLIM-QN does not improve the convergence rate compared to other SQN methods. Our main goal was to show that SLIM-QN practically pushes the boundary of QN methods on large-scale problems while enjoying similar theoretical guarantees.
>
> - No rigorous study to show that the Hessian approximation of the proposed method is more stable than other QN methods.
>
> In Sec 3.1, we use a simple quadratic problem to explain the difference between SLIM-QN and BFGS. Specifically, we analyze the variance of $y_k$ (Table 1) from the classical BFGS and BFGS with momentum. Variance of $y_k$ using momentum is significantly reduced. However, as the reviewer suggested, we will enhance such analysis on the whole Hessian approximation.
>
> - Momentum and damping techniques do not lead to the reduction of compute and memory cost.
>
> Yes this is correct, when compared to the classical L-BFGS. However, we would like to emphasize that the classical L-BFGS is not the baseline we compare to since it cannot be directly used in stochastic NN optimization due to convergence instability. When discussing the reduction of compute cost, we use L-BFGS variants [1,2] as baselines, where they use more costly operations to stabilize the optimization. We will modify the corresponding section to clarify such misunderstandings (in red).
>
> [1] Moritz, P. et al. A linearly-convergent stochastic L-BFGS algorithm. In Artificial Intelligence and Statistics, 2016.
>
> [2] Chang, D. et al. An accelerated linearly convergent stochastic L-BFGS algorithm. IEEE transactions on neural networks and learning systems, 2019.
>
> [3] Ma, S. et al. The power of interpolation: Understanding the effectiveness of SGD in modern over-parameterized learning. In the International Conference on Machine Learning,  2018.
>
> [4] Vaswani, S. et al. Fast and faster convergence of sgd for over-parameterized models and an accelerated perceptron. In The 22nd International Conference on Artificial Intelligence and Statistics 2019.
>
> [5] Oymak, S. et al. Overparameterized nonlinear learning: Gradient descent takes the shortest path?. In the International Conference on Machine Learning 2019.

---

### Decision · Program_Chairs · 2022-01-20

**Decision:**

Reject

**Comment:**

Although the reviewers acknowledge that the paper is well-written and easy to follow, they found that the contributions of the paper are not enough to be accepted at ICLR. Some concerns from the reviewers are as follows:

1. Assumption 3 is very strong and uncommon. It is not easy to verified even for over-parameterized setting.
2. Both the theoretical and experimental results are not sufficient. No improvement in theoretical results compared to the previous work. Moreover, the performance of the method is no better than the baselines, which are themselves much weaker than state-of-the-art results.
3. Motivation for small-batch training, advantages over K-FAC, the practicality of SLIM-QN, and novelty compared to L-BFGS are questionable.
4. The method is essentially LBFGS with momentum and damping of the hessian, hence its novelty is questionable.
5. The authors emphasize that "we are trying to design a practical QN method with light compute/memory cost, especially when applied to large-scale NNs". Any method that has 20-40 times as much memory requirement as SGD cannot be said to have light memory cost.

Based on the above concerns, the paper is not ready for the publication at this moment. The authors should consider to improve the paper by addressing the reviewers' comments and implementing their suggestions and resubmit this paper in the future venues.